# Anatomical and physiological foundations of cerebello-hippocampal interaction

**Thomas Charles Watson[1†‡§#], Pauline Obiang[1†], Arturo Torres-Herraez[1†], Aurélie Watilliaux[1], Patrice Coulon[2], Christelle Rochefort[1], Laure Rondi-Reig[1*]**

[1]Neuroscience Paris Seine, Cerebellum, Navigation and Memory Team, CNRS UMR 8246, INSERM, UMR-S 1130, Sorbonne Universités, University Pierre and Marie Curie, Paris, France; [2]Institut de Neurosciences de la Timone, CNRS and Aix Marseille Université, Marseille, France

**Abstract** Multiple lines of evidence suggest that functionally intact cerebello-hippocampal interactions are required for appropriate spatial processing. However, how the cerebellum anatomically and physiologically engages with the hippocampus to sustain such communication remains unknown. Using rabies virus as a retrograde transneuronal tracer in mice, we reveal that the dorsal hippocampus receives input from topographically restricted and disparate regions of the cerebellum. By simultaneously recording local field potential from both the dorsal hippocampus and anatomically connected cerebellar regions, we additionally suggest that the two structures interact, in a behaviorally dynamic manner, through subregion-specific synchronization of neuronal oscillations in the 6–12 Hz frequency range. Together, these results reveal a novel neural network macro-architecture through which we can understand how a brain region classically associated with motor control, the cerebellum, may influence hippocampal neuronal activity and related functions, such as spatial navigation.
DOI: https://doi.org/10.7554/eLife.41896.001

**\*For correspondence:**
laure.rondi-reig@upmc.fr

[†]These authors contributed equally to this work

**Present address:** [‡]Centre for Discovery Brain Sciences, University of Edinburgh, Edinburgh, United Kingdom; [§]SimonsInitiative for the Developing Brain, University of Edinburgh, Edinburgh, United Kingdom; [#]Patrick Wild Centre for Autism Research, University of Edinburgh, Edinburgh, United Kingdom

**Competing interests:** The authors declare that no competing interests exist.

## Introduction

The cerebellum is classically associated with motor control. However, accumulating evidence suggests its functions may extend to cognitive processes including navigation (*Petrosini et al., 1998*; *Burguière et al., 2005*; *Rondi-Reig and Burguière, 2005*; *Buckner, 2013*; *Koziol et al., 2014*; *Stoodley et al., 2017*). Indeed, anatomical and functional connectivity has been described between cerebellum and cortical areas that are engaged in cognitive tasks (*Kim et al., 1994*; *Kelly and Strick, 2003*; *Ramnani, 2006*; *Watson et al., 2009*; *Watson et al., 2014*; *Stoodley and Schmahmann, 2010*). Furthermore, the cerebellum has recently been found to form functional networks with subcortical structures associated with higher order functions, such as the basal ganglia (*Kelly and Strick, 2004*; *Hoshi et al., 2005*; *Bostan et al., 2010*; *Chen et al., 2014*; see *Bostan and Strick, 2018* for review), ventral tegmental area (*Rogers et al., 2011*; *Carta et al., 2019*) and hippocampus (*Rochefort et al., 2013*; *Iglói et al., 2015*; *Yu and Krook-Magnuson, 2015*; *Babayan et al., 2017*).

In the hippocampus, spontaneous local field potential (LFP) activity (*Iwata and Snider, 1959*; *Babb et al., 1974*; *Snider and Maiti, 1975*; *Krook-Magnuson et al., 2014*) and place cell properties (*Rochefort et al., 2011*; *Lefort et al., 2019*) are profoundly modulated following cerebellar manipulation (*Rondi-Reig et al., 2014* for review). A recent study has also described, at the single cell and blood-oxygen-level-dependent signal level, sustained activation in the dorsal hippocampus during optogenetic enhancement of cerebellar nuclei output in head-fixed mice (*Choe et al., 2018*). These data point toward the existence of an anatomical projection from the cerebellum to the hippocampus. However, the direct or indirect nature of the connection between the two structures remains unclear. The suggestion of a direct connection between these two structures has been claimed by a

recent tractography study in humans (*Arrigo et al., 2014*) and the presence of short-latency evoked field potentials (2–4 ms) in cat and rat hippocampi after electrical stimulation of the cerebellar vermal and paravermal regions (*Whiteside and Snider, 1953*; *Harper and Heath, 1973*; *Snider and Maiti, 1976*; *Heath et al., 1978*; *Newman and Reza, 1979*). However, secondary hippocampal field responses have also been described, at a latency of 12–15 ms following cerebellar stimulation, suggesting the existence of an indirect pathway (*Whiteside and Snider, 1953*).

Taken together, even if these studies provide compelling physiological evidence of cerebellar influences on the hippocampus, they do not provide direct evidence of neuroanatomical connectivity between the two regions. Given the known complex, modular functional and anatomical organization of the cerebellum (*Apps and Hawkes, 2009*) this represents a major gap in our understanding of the network architecture linking the two structures. In addition, these studies provide no measure of dynamic physiological communication or associated synchronization between the two structures, which is thought to be essential for maintaining distributed network functions (e.g. *Fries, 2005*).

Therefore, this study addresses two fundamental, unanswered questions: which regions of the cerebellum are anatomically connected to the hippocampus and what are the spatio-temporal dynamics of synchronized cerebello-hippocampal activity during behavior? To address these unresolved questions, we used rabies virus as a retrograde transneuronal tracer to determine the extent and topographic organization of cerebellar input to the hippocampus. Based upon the anatomical tracing results, we then studied the levels of synchronization between LFPs recorded simultaneously from the cerebellum and dorsal hippocampus in freely moving mice as a proxy for potential cross-structure interactions. We reveal that specific cerebellar modules are anatomically connected to the hippocampus and that these inter-connected regions dynamically synchronized their activity during behavior.

## Results

To study the topographical organization of ascending, cerebello-hippocampal projections, we unilaterally injected rabies virus (RABV), together with fluorescent cholera toxin β-subunit (CTb), into the left hippocampus. The extent of the injection site, identified by the presence of fluorescent CTb, included both CA1 (stratum lacunosum-moleculare) and DG (molecular layer, granule cell layer and hilus, *Figure 1—figure supplement 1*).

### Cerebellar projections to the hippocampus are precisely topographically organized

We characterized the presence of retrograde transneuronally RABV-infected neurons after survival times of 30, 48, 58 and 66 hr (*Suzuki et al., 2012*; *Aoki et al., 2017*; *Coulon et al., 2011*). At 30 hr post-infection, staining was found predominantly in the medial septum diagonal band of Broca (MSDB), the lateral and medial entorhinal cortices, the perirhinal cortex and the supramammillary nucleus of the hypothalamus (SUM). Sparse staining was also found in a subset of mice, in the raphe nucleus and in other hypothalamic and thalamic regions (*Figure 1—figure supplement 2*). All these structures correspond to first-order relays as CTb staining was also systematically associated with each of the sites containing rabies infected neurons (*Figure 1—figure supplement 3*). Importantly, no RABV or CTb labeling was found in the cerebellum at this time-course ruling out the existence of a direct cerebello-hippocampal DG pathway in mice. Interestingly, among these putative first relays, the septum, the hypothalamus and the raphe nucleus receive direct projections from the deep cerebellar nuclei in cat (*Paul et al., 1973*), rat (*Teune et al., 2000*) and Macaca (*Haines et al., 1990*). In accordance with this, some RABV-infected neurons were observed in the deep cerebellar and vestibular nuclei 48 hr post-hippocampal infection and some labeled cells were found in the cerebellar cortex at 58 hr p.i. (*Figure 1A*, inset, *Figure 1—figure supplement 4*) suggesting the existence of single-relay pathways (i.e. di-synaptic connections) between the cerebellum and hippocampus.

At 48 hr post-infection, RABV labeling was also found in mammillary bodies, amygdala and several midbrain and pontine regions such as the periaqueductal gray (PAG), the nucleus incertus and the laterodorsal tegmental nucleus (LtDG). These regions, which are known to receive direct projections from the DCN (*Teune et al., 2000*) were neither stained with CTb nor RABV-positive at 30 hr post-infection and therefore represent a putative additional pathway involving two relays between the cerebellum and the hippocampus (*Figure 1—figure supplement 4*).

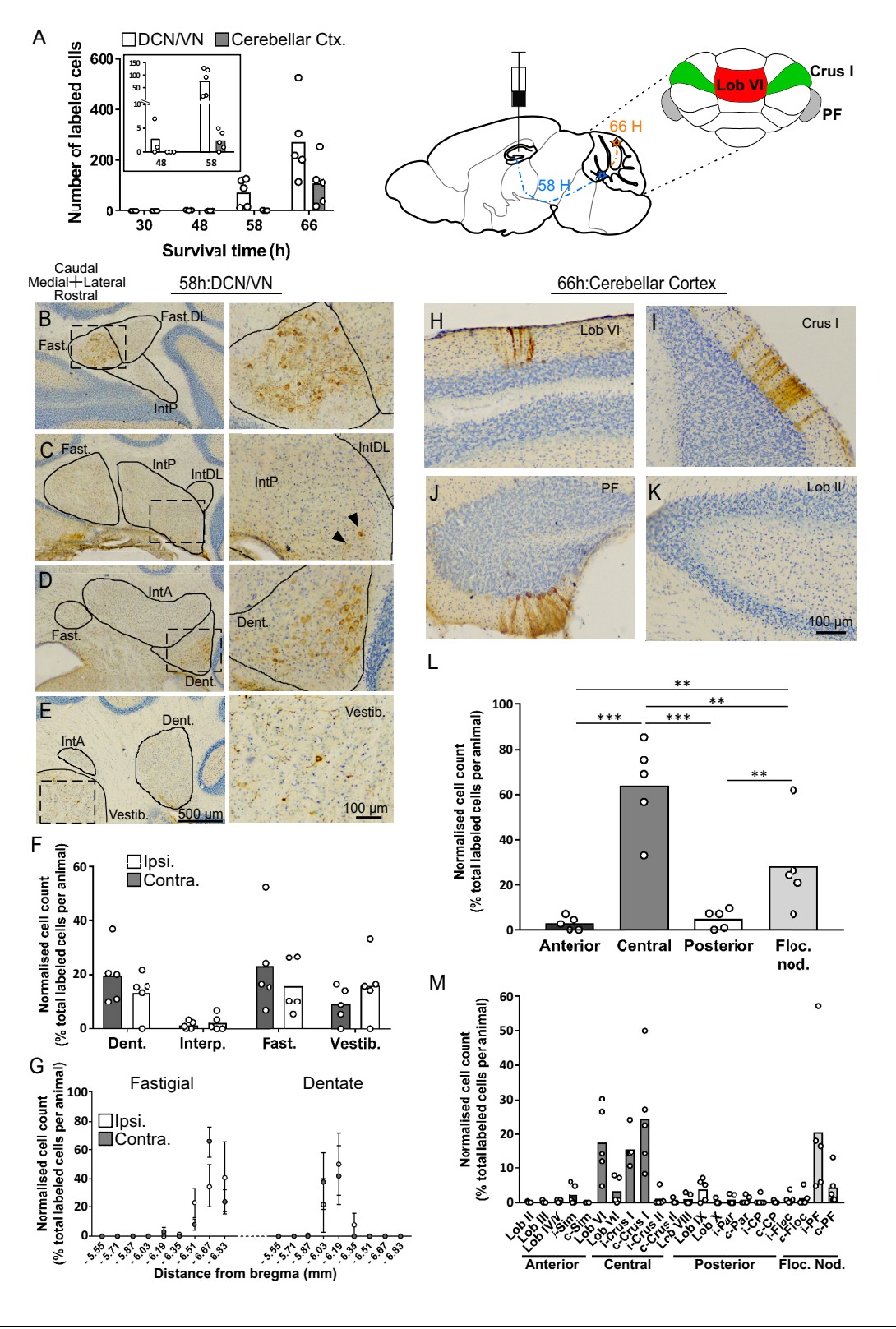

**Figure 1.** Topographically restricted regions of cerebellar cortex are connected to the hippocampus. (A) Left, mean number of labeled cells in the deep cerebellar nuclei (DCN), vestibular nuclei (VN) and cerebellar cortex at different survival times following rabies injection in left hippocampal dentate gyrus. Box shows a magnification of the labeling at 48 and 58 hr (n = 5 mice for 58 hr and 66 hr, 30 hr, n = 4 mice and 48 hr, n = 3 mice). Middle, schematic representation of rabies injection site and survival times required to reach the cerebellar and vestibular nuclei (58 hr, dashed blue

*Figure 1 continued on next page*

*Figure 1 continued*

line), and cerebellar cortex (66 hr, orange line). Upper right, schematic view of the posterior cerebellar cortex indicating regions of highest labeling following rabies virus injection (red, vermis lobule VI; green, Crus I; gray, paraflocculus). (**B-E**) Representative photomicrographs showing labeling in the cerebellar and vestibular nuclei 58 hr post-infection. Left panels show low-magnification view, right panels show magnified view of area indicated by dashed box. Solid arrow heads indicate the presence of the very few labeled cells in the IntP. (**F**) Pooled, normalized counts of rabies labeled cells in the ipsi- and contralateral cerebellar and vestibular nuclei 58 hr post-infection (n = 5 mice). No significant differences were found between ipsi- and contralateral nuclei (nuclei x hemisphere two-way ANOVA, hemisphere effect $F_{(1, 4)}=1.31\times10^{-5}$, p=0.99, interaction effect $F_{(3, 12)}=2.79$, p=0.09, nuclei effect $F_{(3, 12)}=9.38$, p=0.002). (**G**) Normalized cell counts in the fastigial nucleus (left) and dentate nucleus (right), according to their rostro-caudal position relative to bregma. Open circles, contralateral count; filled circles, ipsilateral count (n = 5 mice). (**H-K**) Representative photomicrographs of the resultant labeling in lobule VI, Crus I, paraflocculus and lobule II at 66 hr post-infection. (**L**) Normalized count of rabies labeled cells in anterior (black bar; lobule II to lobule IV/V); central (dark gray bar; lobule VI to Crus II); posterior (clear bar; lobule VIII and lobule IX) and flocculonodular (Floc. Nod., light gray bar; lobule X, flocculus and paraflocculus) cerebellum 66 hr post-infection (n = 5 mice; one-way ANOVA with FDR correction, $F_{(3, 16)}=19.11$, p<0.0001). (**M**) Normalized cell count of rabies labeled cells in all assessed lobules 66 hr post-infection. Color coding of bars indicate assignment of lobules to either anterior, central, posterior or vestibular cerebellum as indicated in L. Abbreviations: Dent., Dentate nucleus; Fast., fastigial nucleus; Fast. DL, dorsolateral fastigial nucleus; Floc. Nod., flocculonodular lobe; Interp., nucleus interpositus; IntA, nucleus interpositus anterior; IntDL, dorsolateral nucleus interpositus; IntP, nucleus interpositus posterior; i-Sim, ipsilateral simplex lobule; c-Sim, contralateral simplex lobule; i-Crus I, ipsilateral Crus I; c-Crus I, contralateral Crus I; i-Crus II, ipsilateral Crus II; c-Crus II, contralateral Crus II; i-Par, ipsilateral paramedian lobule; c-Par, contralateral paramedian lobule; i-CP, ipsilateral copula; c-CP, contralateral copula; i-Floc, ipsilateral flocculus; c-Floc, contralateral flocculus; i-PF, ipsilateral paraflocculus; c-PF, contralateral paraflocculus; Vestib., vestibular nuclei. ** q < 0.01, *** q < 0.001.

DOI: https://doi.org/10.7554/eLife.41896.002

The following source data and figure supplements are available for figure 1:

**Source data 1.** Summary table of RABV labeling in the cerebellum 58 hr post hippocampal infection.
DOI: https://doi.org/10.7554/eLife.41896.008
**Figure supplement 1.** Location of rabies virus injections for the four experimental groups.
DOI: https://doi.org/10.7554/eLife.41896.003
**Figure supplement 2.** Structures labeled 30 hr after hippocampal rabies injection.
DOI: https://doi.org/10.7554/eLife.41896.004
**Figure supplement 3.** Analysis of CTB retrograde labeling indicates that rabies labeled structures at 30 hr post-infection are potential first relay regions.
DOI: https://doi.org/10.7554/eLife.41896.005
**Figure supplement 4.** Summary table of RABV labeling across brain structures 48 hr after hippocampal rabies injection Overview of RABV labeling in different brain regions 48 hr post rabies injection in the hippocampus.
DOI: https://doi.org/10.7554/eLife.41896.006
**Figure supplement 5.** The topographical distribution of DCN labeling at 66 hr.
DOI: https://doi.org/10.7554/eLife.41896.007

58 hr post-infection, abundant and strong staining was found in the fastigial and dentate nuclei, with almost no staining in the interpositus (*Figure 1B–F*). Within the fastigial and dentate, RABV-labeled cells were found to be topographically restricted to caudal and central regions, respectively (*Figure 1G*).

Following 66 hr of incubation, the number of strongly labeled cells increased in the DCN and vestibular nuclei (*Figure 1A*); however, the topographical distribution remained unchanged (*Figure 1—figure supplement 5*). At the level of the cerebellar cortex, longitudinal clusters of RABV+ Purkinje cells (PCs) were found bilaterally across highly restricted central and flocculo-nodular regions (*Figure 1L*). In the central cerebellum (i.e. lobules VI, VII and associated hemispheres), clusters were particularly concentrated in lobule VI and Crus I (*Figure 1H,I,M*). In the flocculo-nodular cerebellum, RABV-labeled cells were found in the dorsal and ventral paraflocculus (*Figure 1J and M* and *Figure 2B*). Within the vermis we identified a single cluster of RABV+ Purkinje cells that extended across both lobule VIa and lobule VIb-c (*Figure 2*). In contrast, within Crus I, RABV-labeled Purkinje cells were arranged in two spatially isolated clusters, one located rostro-laterally and the other caudo-medially (*Figure 2*).

The topographical arrangement of RABV-labeled PCs in longitudinal clusters is in keeping with the well-described modular organization of the cerebellum (e.g. *Apps and Hawkes, 2009*). Mapping of molecular marker expression patterns, such as zebrin II banding, provides a reliable basis from which modules can be defined and recognized in the cerebellar cortex of rodents. Thus, to further assign the observed PC clusters to previously described cerebellar zones, we used a double

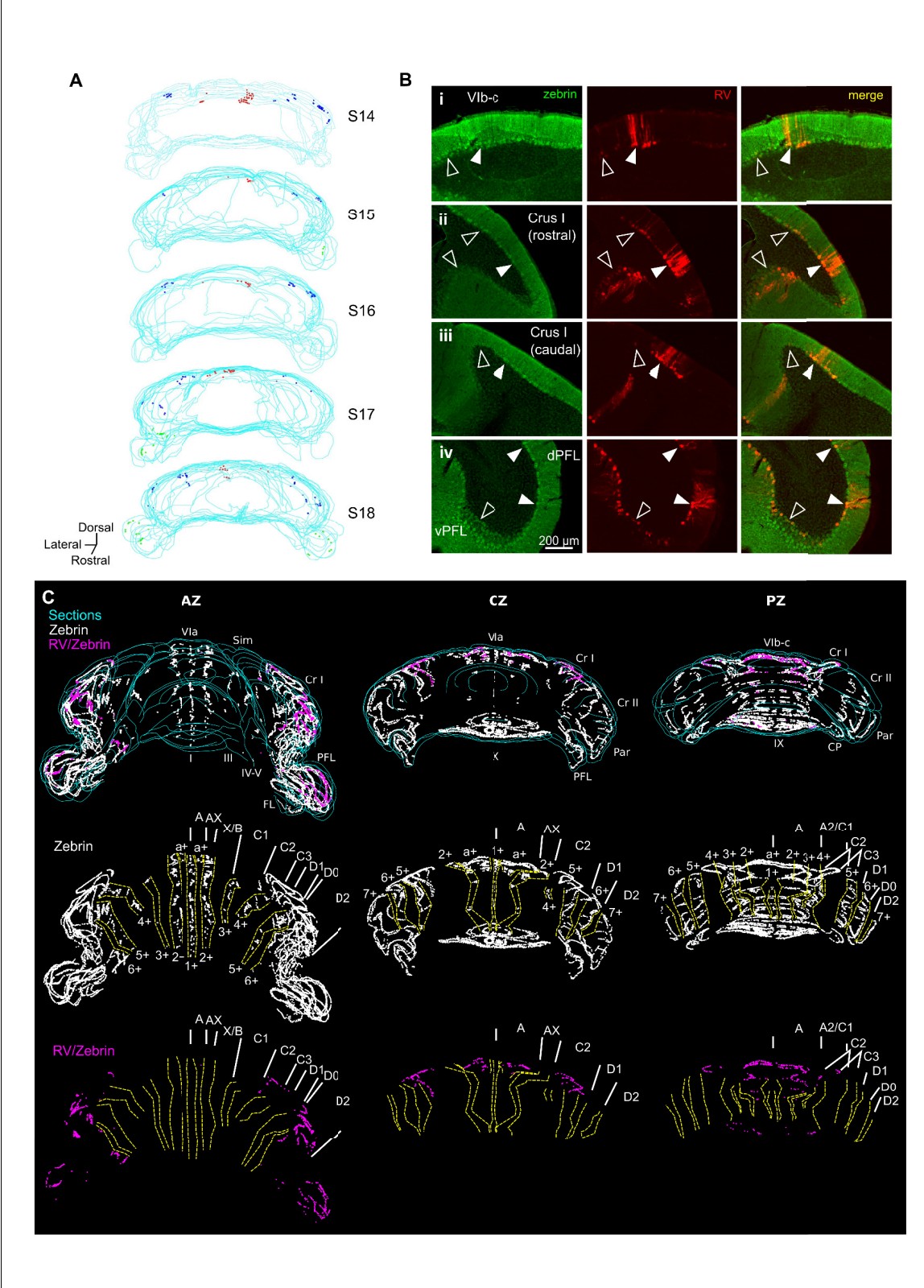

**Figure 2.** Different cerebellar modules project to the hippocampus. (**A**) 3-D reconstruction showing the location of RABV+ Purkinje cells in the most labeled cerebellar lobules at 66 hr post-infection. Red, blue and green dots represent RABV+ Purkinje cells in lobule VI, Crus I and paraflocculus, respectively. (**B**) Photomicrographs from case S18 showing double staining against zebrin II (green, left column), RABV (red, central column) and merge (right column) in lobule VI (i), Crus I (ii and iii) and paraflocculus (iv). RABV+ Purkinje cells were also zebrin positive and were organized in clusters of

*Figure 2 continued on next page*

Figure 2 continued

strongly labeled RABV+ cells (filled arrow-heads) surrounded by weakly labeled RABV+ Purkinje cells (unfilled arrow-heads). (C) Assignment of the RABV + clusters to specific cerebellar modules for case S18 in the anterior (AZ; left), central (CZ; central column) and posterior (PZ; right column) zones. First row shows stacked sections with zebrin-positive Purkinje cells (white dots) and RABV+ Purkinje cells, which were also zebrin positive (purple dots, strong and weakly labeled cells included); central row shows reconstructed principal zebrin bands (delineated by yellow dashed lines and named from 1+ to 7 +; nomenclature from *Sugihara and Quy, 2007*) and cerebellar modules (capital letters; defined as in *Sugihara and Quy, 2007*); and bottom row shows the location of the RABV+/zebrin Purkinje cells (purple dots) in relation to reconstructed zebrin bands and modules. Abbreviations, I, lobule I; III, lobule III; IV/V, lobule IV/V; VIa and VI b-c, lobule VIa and VI b-c, respectively; IX, lobule IX; X, lobule X; Sim, lobule simplex; Cr I, Crus I; Cr II, Crus II; Par, paramedian lobule; CP, copula, PFL, paraflocculus, FL, flocculus.; dPFC and vPFC, dorsal and ventral paraflocculus, respectively.

DOI: https://doi.org/10.7554/eLife.41896.009

immunohistochemical approach to stain for both RABV and aldolase C (zebrin II) in one animal (case S18) after 66 hr of infection (*Figure 2B*) (*Brochu et al., 1990*; *Sugihara and Shinoda, 2004*). Lobule VI, Crus I and paraflocculus are mostly zebrin-positive regions (*Sugihara and Quy, 2007*), and we found that RABV-labeled Purkinje cells co-localized with zebrin II in all the observed clusters (*Figure 2B*). In the vermis, lobule VIa RABV-labeled PCs were mostly located in the a+ band. The few RABV-labeled cells found in lobule VII were confined to the 2+ band. Thus, together, these labeled cells belong to the a+//2+ pair that constitutes part of the cerebellar A module (*Figure 2C*) (*Sugihara, 2011*). In Crus I, the rostrolateral cluster of RABV-labeled PCs was aligned with the anterior 6+ zebrin band corresponding to module D2. The caudomedial cluster was in continuation with the posterior 5+ zebrin band suggesting that it is part of the paravermal module C2 (*Figure 2C*). In the paraflocculus, the assignment of the RABV-labeled cells to specific modules was not addressed given the complex morphology of this region. However, the presence of RABV-labeled cells both in the dorsal and ventral paraflocculus suggests the involvement of more than one module (*Figure 2B–C*) (*Voogd and Barmack, 2006*).

Cerebellar modules are also defined by their outputs through the deep cerebellar and vestibular nuclei (*Apps and Hawkes, 2009*; *Ruigrok, 2011*). The presence of RABV-labeled cells in the fastigial nucleus is consistent with the involvement of module A. Similarly, the D2 module is routed through the dentate nuclei in which we found robust RABV labeling. We also found RABV+ cells in the nucleus interpositus posterior, which provides the output of module C2. Finally, RABV labeling was observed in the vestibular nuclei, which may represent the output of RABV+ Purkinje cell clusters observed in the ventral paraflocculus.

Together, our neuroanatomical tracing data indicate that cerebellar projections to the hippocampus emanate from three distinct cerebellar modules. It also suggests the existence of multiple, convergent pathways from the DCN to the hippocampus.

## Cerebello-hippocampal physiological interactions in a familiar home-cage environment

In order to question the potential functional relevance of cerebello-hippocampal anatomical connectivity, we implanted mice (n = 21) with arrays of bipolar LFP recording electrodes in bilateral dorsal hippocampus (HPC) and unilaterally in two highly RABV-labeled regions of the central cerebellum, lobule VI (midline) and Crus I (left hemisphere). For comparison, we also simultaneously recorded LFP from cerebellar regions with minimal RABV labeling (lobule II or lobule III; *Figure 1M*; *Figure 3A and B*). Data were excluded from further analysis in cases where postmortem histological inspection revealed that electrode positions were off-target (*Figure 3—figure supplement 1*).

The spectral profile of cerebellar and hippocampal LFP activity was first assessed during active movement in a familiar home-cage environment (epochs with speed >3 cm/s; see Materials and methods). Within the HPC, a dominant 6–12 Hz theta oscillation was similarly observed in both hemispheres (*Figure 3—figure supplement 2B*; left HPC: mean 6–12 Hz z-score power = 4.17 ± 0.20, n = 17; right HPC: mean 6–12 Hz z-score power = 4.52 ± 0.16, n = 19; unpaired t test, $t_{34}$ = 0.007, p=0.1731), thus, when LFPs from both hippocampi where available we averaged the spectral power from left and right hemispheres (*Figure 3C*, mean 6–12 Hz z-score power = 4.35 ± 0.14). In the cerebellar recordings, low-frequency oscillations in the delta (2–4 Hz) and high-frequency activity in the mid-high gamma (50–140 Hz) range were prevalent. A clear peak in the 6–12 Hz band was not detected in the power spectrum of any of these recordings, which had

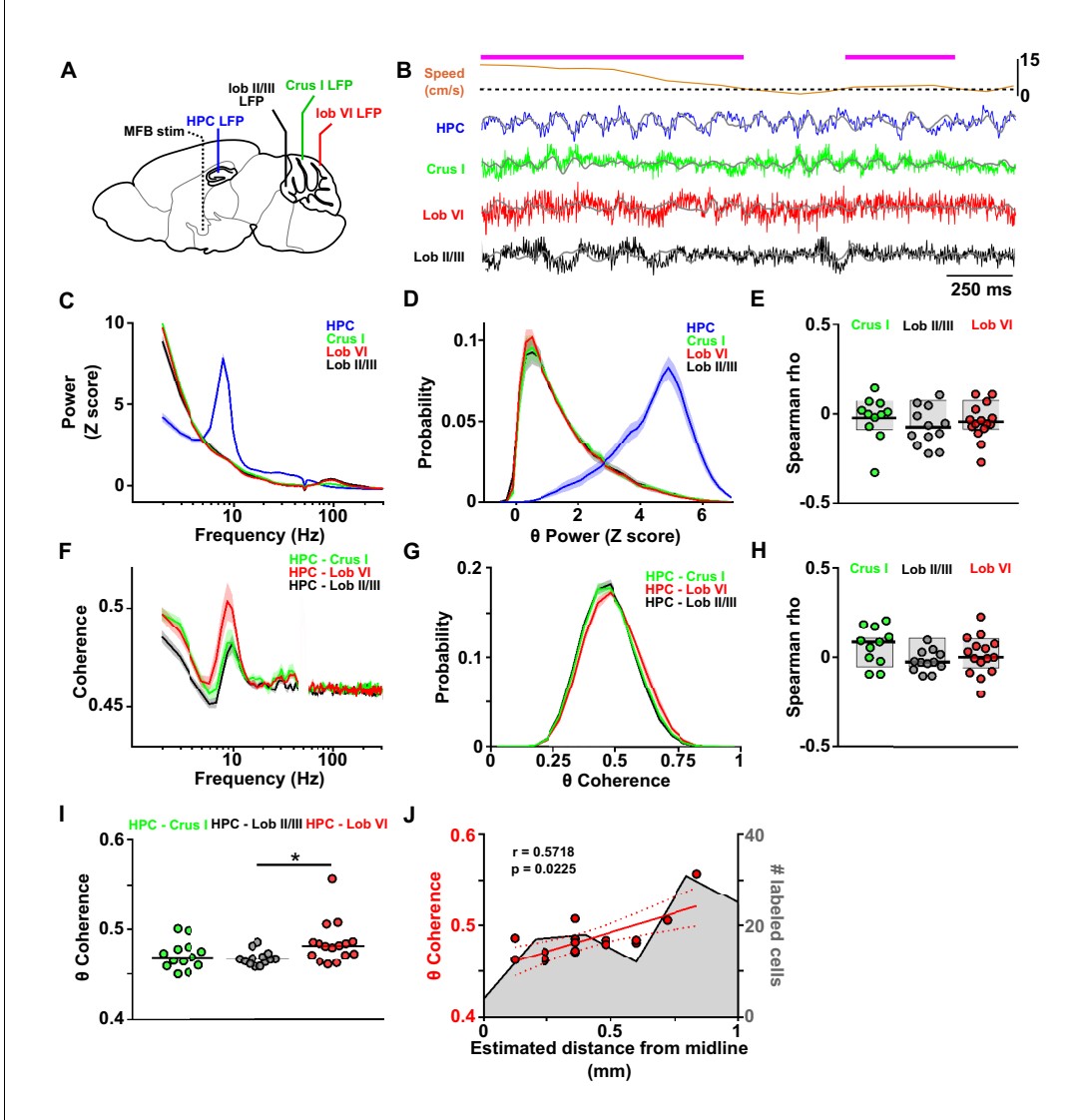

**Figure 3.** Assessment of cerebello-hippocampal interactions during active movement in the homecage. (A) Schematic representation of recording and stimulation electrode implant positions. (B) Representative simultaneous raw LFP recordings (colored lines) and LFP filtered in the theta frequency range (θ, 6–12 Hz, overlaid gray lines) recorded during active movement in the homecage condition as defined by instantaneous speed. Solid magenta line indicates data epochs in which speed was above the required threshold for inclusion in further analysis (3 cm/s, dashed line (20 mice). In one mouse in which speed data was not available we used neck electromyograph (EMG) data to define periods of active movement). (C) Averaged z-score of the power spectra from hippocampal LFP (n = 21, mean between left and right hemisphere LFPs when both recording electrodes were on target) and cerebellar cortical regions Crus I (n = 12), lobule II/III (n = 13) and lobule VI (n = 16) during homecage exploration (speed above 3 cm/s). (D) Probability distribution of the instantaneous z-scored theta power for each of the recorded regions. Hippocampal theta power followed a negatively skewed distribution while cerebellar theta power followed a positively skewed distribution. (E) Correlation between the instantaneous z-scored theta power for each of the cerebellar recorded regions and instantaneous speed. Horizontal line indicates mean. Gray-shaded bars represent the confidence levels obtained from bootstrapped data with α = 0.05. Theta power was not significantly correlated with speed in any of the cerebellar recordings (Crus I, n = 11; lobule II/III, n = 12; lobule VI, n = 15). (F) Averaged coherence between cerebellar cortical recordings (color coded) and hippocampus (when both hippocampal recording electrodes were on target we averaged the coherence obtained with left and right hemispheres; Crus I, n = 12; lobule II/III, n = 13; lobule VI, n = 16) during homecage exploration (speed above 3 cm/s). (G) Probability distribution of the instantaneous hippocampal-cerebellar theta coherence (color coded). All the recording combinations followed normal distributions. (H) Correlation between the instantaneous hippocampal-cerebellar theta coherence (color coded) and the instantaneous speed. Horizontal line indicates mean. Gray-shaded bars represent the confidence levels obtained from bootstrapped data with α = 0.05. HPC-cerebellar theta coherence was not significantly correlated with speed in any of the recording combinations. (I) Theta coherence between cerebellar recordings and hippocampus (average between coherence with left and right hemisphere LFPs when both recordings were on target; Crus I - HPC, n = 12; lobule II/III - HPC, n = 13; lobule VI - HPC, n = 16). Lobule VI-HPC coherence was significantly higher than that observed with lobule II/III (horizontal line indicates median coherence for each combination; *, Kruskal-

*Figure 3 continued on next page*

*Figure 3 continued*

Wallis with FDR correction, Kruskal-Wallis statistic = 6.75, p = 0.0342; lobule VI-HPC vs lobule II/III-HPC, p = 0.0246). J, Mean lobule VI-HPC theta coherence plotted against the estimated medio-lateral position of the recording electrode in lobule VI (red dots; n = 16, when both hippocampal recording electrodes were on target we averaged the coherence obtained with left and right hemispheres; Spearman rho = 0.5718, p = 0.0225). In gray, number of RABV+ cells counted across lobule VI 66 hr after injection in the left HPC as a function of medio-lateral position (0.2 mm bins; n = 5 mice). Shading indicates S.E.M. Abbreviations, LFP, local field potential; HPC, dorsal hippocampus; lob II/III, lobule II/III, lob VI, lobule VI, MFB stim, medial forebrain bundle stimulation.

DOI: https://doi.org/10.7554/eLife.41896.010

The following source data and figure supplements are available for figure 3:

**Source data 1.** Assessment of cerebello-hippocampal interactions during active movement in the homecage.

DOI: https://doi.org/10.7554/eLife.41896.015

**Figure supplement 1.** Reconstructed location of the implanted electrodes.

DOI: https://doi.org/10.7554/eLife.41896.011

**Figure supplement 2.** Cerebello-hippocampal coherence patterns are similar across hemispheres during active movement in homecage.

DOI: https://doi.org/10.7554/eLife.41896.012

**Figure supplement 3.** Experimental setup for simultaneous recording of photo-identified Purkinje cells and hippocampal local field potential.

DOI: https://doi.org/10.7554/eLife.41896.013

**Figure supplement 4.** Polar plots and associated histograms of cerebellar units significantly phase locked to hippocampal 6–12 Hz oscillations.

DOI: https://doi.org/10.7554/eLife.41896.014

similar levels of 6–12 Hz z-score power (*Figure 3C*; Crus I: 1.57 ± 0.10, n = 12 mice; lobule II/III: 1.57 ± 0.13, n = 13 mice; lobule VI: 1.47 ± 0.07, n = 16 mice; one-way ANOVA, $F_{(2, 38)}$=0.34, p=0.7131).

In order to explore any potential state-dependent organization of theta oscillations in our recordings, we computed the distributions of instantaneous power in this frequency range (*Figure 3D*). If the power of theta oscillations was organized in differential states, we would expect to observe a multimodal distribution; however, we found unimodal power distributions in both the hippocampal and cerebellar recordings. The former was negatively skewed suggesting that the peak observed in the hippocampal power spectra is representative of sustained theta activity in the analyzed epochs. In contrast, the cerebellar theta distribution profile was positively skewed suggesting that activity in this frequency range, although of lower power than in the hippocampus, is also sustained within the cerebellar cortical regions we recorded from.

Previous studies have described correlations between locomotion speed and purkinje cell discharge in cerebellar vermal lobules V and VI (*Sauerbrei et al., 2015*; *Muzzu et al., 2018*). Furthermore, cerebellar nuclei - prefrontal cortex theta coherence has also been found to increase during active locomotion compared to rest (*Watson et al., 2014*). Therefore, we next asked if running speed could be modulating theta power in the cerebellar cortex. To do so, we computed the correlation between instantaneous speed and instantaneous theta power in each cerebellar region; however, none were significantly correlated (*Figure 3E*; Crus I, n = 11, mean Spearman rho = - 0.02 ± 0.04, bootstrap confidence level = [- 0.09 0.08]; lobule II/III, mean Spearman rho = - 0.07 ± 0.03, bootstrap confidence level = [- 0.08 0.08]; lobule VI, mean Spearman rho = - 0.04 ± 0.03, bootstrap confidence level = [- 0.08 0.08]).

As an indicator of cross-structure interaction (*Fries, 2005*), we next calculated coherence between LFP recorded from the different cerebellar subregions and left or right HPC. We found no statistically significant influence of hippocampal laterality on the measured cerebello-hippocampal coherence (*Figure 3—figure supplement 2*; Crus I-HPC left, n = 11, Crus I-HPC right, n = 11, Mann-Whitney test, U = 59, p=0.9487; lobule II/III-HPC left, n = 11, lobule II/III-HPC right, n = 12, Mann-Whitney test, U = 63, p=0.8801; lobule VI-HPC left, n = 15, lobule VI – HPC right, n = 14, Mann-Whitney test, U = 105, p>0.99). Therefore, for further analysis, when both hippocampal recording electrodes were on target we first calculated coherence with the cerebellar LFP for each hemisphere then averaged the two. Thus, for each mouse we obtained one coherence value per cerebello-hippocampal recording combination. In cases in which only one of the hippocampal recording electrodes was on target, we excluded the off target recording in our calculations of cerebello-hippocampal coherence.

A clear peak in coherence was observed for all cerebello-hippocampal combinations in the theta frequency range (6–12 Hz, *Figure 3F*; Crus I-HPC, n = 12; lobule II/III-HPC, n = 13, lobule VI-HPC, n = 16; frequencies between 48 and 52 Hz have been excluded due to notch filtering to remove electrical contamination of the LFP signals; see Materials and methods). We found that significant variations in coherence level were restricted to those within the theta frequency (frequency band (theta, 6–12 Hz; beta, 13–29 Hz; low gamma, 30–48 Hz) x cerebello-hippocampal combination two way repeated measures ANOVA, frequency band effect $F_{2,76}$ = 22.42, p<0.0001; combination effect $F_{2,38}$ = 2.843, p=0.0707; interaction effect, $F_{4,76}$ = 3.825, p=0.0069; post-hoc multiple comparisons with FDR correction revealed significant differences between combinations only in the theta band). In addition, as theta coherence has already been reported as a potential mechanism for long-range network interactions between the hippocampus and the cerebellum (*Hoffmann and Berry, 2009*; *Wikgren et al., 2010*) and theta oscillations are known to play an important role in intra-hippocampal network organization, in particular for spatial navigation (see *Buzsáki and Moser, 2013* for overview), we focused our analysis on activity within this frequency range. As for the instantaneous LFP power, we next asked if theta coherence between the different cerebello-hippocampal LFP combinations was organized in a state-dependent manner. To address this question, we computed the distribution of instantaneous coherence within this bandwidth. In line with the analysis of LFP power distributions, we did not observe any multi-modality and all combinations followed gaussian distributions (*Figure 3G*). Correlation analysis between instantaneous speed and instantaneous theta coherence failed to show significant relationships for any cerebello-hippocampal combination (*Figure 3H*; Crus I-HPC, mean Spearman rho = 0.06 ± 0.03, bootstrap confidence level = [−0.08 0.08]; lobule II/III-HPC, mean Spearman rho = - 0.02 ± 0.02, bootstrap confidence level = [- 0.08 0.08]; lobule VI-HPC, mean Spearman rho = 0.01 ± 0.03, bootstrap confidence level = [−0.09 0.08]). Significant differences across recording combinations were observed within the theta bandwidth (*Figure 3I*; Crus I-HPC, median [25–75 interquartile range (IQR)] coherence = 0.471 [0.461–0.480]; lobule II/III-HPC, median [IQR] coherence = 0.467 [0.462–0.471]; lobule VI-HPC, median [IQR] coherence = 0.481 [0.471–0.486]; Kruskal-Wallis with FDR correction, Kruskal-Wallis statistic = 6.75, p=0.0342) and post-hoc analysis revealed that LFP oscillations were significantly more synchronized between hippocampus and lobule VI than with lobule II/III (corrected p = 0.0246). Within lobule VI, theta coherence was significantly correlated to the mediolateral position of the recording electrode, which was consistent with the mediolateral location of greatest RABV-labeled PCs (*Figure 3J*; Spearman rho = 0.572, p=0.0225).

Next, to ascertain if local cerebellar spiking activity is coordinated or modulated by hippocampal theta oscillations we recorded single, photo-identified Purkinje cells from lobule VI of head-fixed L7-ChR2 mice (selectively expressing channelrhodopsin in Purkinje cells; n = 6) simultaneously with hippocampal LFP during periods of active movement. This allowed us to calculate the degree of phase-locking between the cerebellar spikes and hippocampal LFP, which circumvents volume conduction issues associated with LFP-LFP correlations (e.g. *Vinck et al., 2011*; *Nolte et al., 2004*; *Sirota et al., 2008*; *Stam et al., 2007*). We recorded a total of 22 units, of which 16 (from four mice) were classified as Purkinje cells based upon their responsivity to blue light illumination (see *Figure 3—figure supplement 3*). Of these 16 Purkinje cells, 31% (5 units) were significantly phase locked to the hippocampal 6–12 Hz oscillation (*Figure 3—figure supplement 4*) during periods of active movement (see Materials and methods). Of the six non-photo responsive units, 1 (16.7%) was significantly phase locked (*Figure 3—figure supplement 4D,E*). The mean vector angle of the significantly phase-locked units was 231 ± 18°.

## Cerebello-hippocampal interactions during the learning of a goal-directed behavior

To further characterize the dynamics of cerebello-hippocampal interactions, we quantified cerebello-hippocampal theta coherence during a goal-directed task. A subset of mice (n = 8) were trained to traverse a linear track to get a reward (medial forebrain bundle stimulation, see Materials and methods) at a fixed position (*Figure 4A*).

Across training, mice improved their performance as shown by the optimization of their path (*Figure 4A*), significant increase in the mean number of rewards obtained per day of training (*Figure 4B*; mean number of rewards obtained on 1[st] day = 17 ± 4, mean number of rewards obtained on 7th day = 68 ± 11; repeated measures Friedman test, Friedman statistic = 37.91,

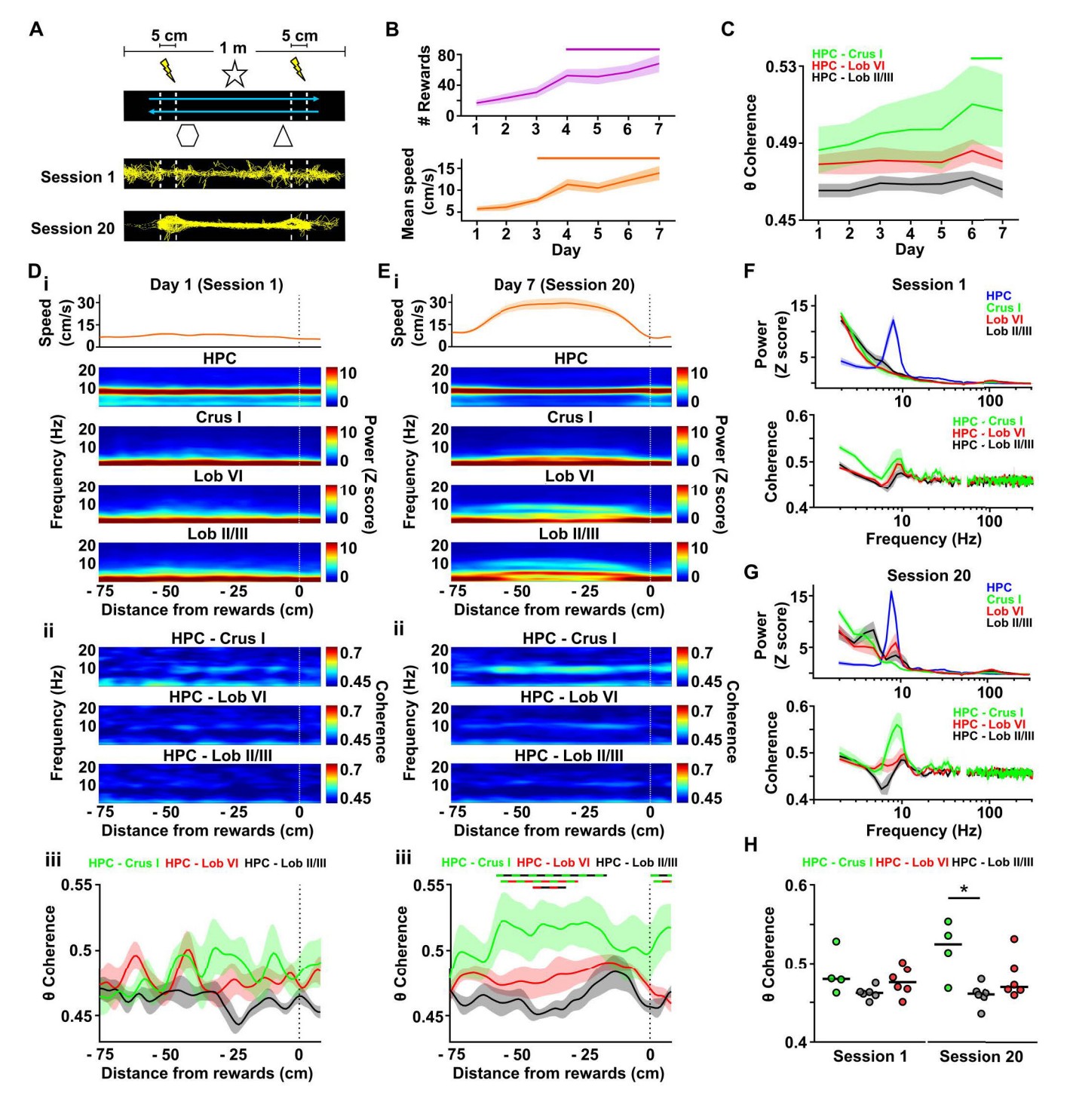

**Figure 4.** Cerebello-hippocampal interactions during goal-directed behavior. (**A**) Mice learned to traverse a 1 m linear track to receive a medial forebrain bundle stimulation (lightening symbols) upon reaching invisible goal zones (n = 8 mice). Representative trajectories from early (session 1) and late (session 20) training show the transition from exploratory to goal-directed behavior. (**B**) Mice improved their performance in the task across days as shown by increases in the mean number of rewards obtained (average of the three sessions per day, repeated measures Friedman test with FDR correction, Friedman statistic = 37.91, p < 0.0001; solid line, day 1 vs days 4–7, p < 0.01) and mean speed (average of the three sessions per day, repeated measures Friedman test with FDR correction, Friedman statistic = 36.32, p < 0.0001; solid line, day 1 vs days 4–7, p < 0.05). (**C**) Overall cerebello-hippocampal theta coherence per day (average of the three sessions per day) during learning of the linear track task (when both hippocampal recording electrodes were on target we averaged the coherence obtained with left and right hemispheres; Crus I, n = 4; lobule II/III, n = 6; lobule VI,

*Figure 4 continued on next page*

*Figure 4 continued*

n = 6). Hippocampus-Crus I coherence increased significantly compared with first day (day of training x cerebello-hippocampal combination two-way ANOVA with FDR correction, day effect $F_{6,84}$ = 3.873, p = 0.0018; solid line, day 1 vs days 6–7, p < 0.01). (D) i, Top: Mean speed aligned by distance from the reward location (position 0) averaged across runs during session 1. Bottom: Mean power spectrogram aligned by distance from reward and averaged across runs during session one for hippocampus LFP (n = 8, mean between left and right hemisphere LFPs when both hippocampal recordings were on target) and cerebellar cortical regions (Crus I, n = 4; lobule II/III, n = 6; lobule VI, n = 6). ii, Mean coherogram aligned by distance from reward location (position 0) averaged across runs during session one for each hippocampal-cerebellar combination (when both hippocampal recording electrodes were on target we averaged the coherence obtained with left and right; Crus I, n = 4; lobule II/III, n = 6; lobule VI, n = 6). iii, Mean theta coherence aligned by distance from reward and averaged across runs during session one for each hippocampal-cerebellar combination (mean between coherence with left and right hemisphere LFPs when both recordings were on target; Crus I, n = 4; lobule II/III, n = 6; lobule VI, n = 6). No significant differences between hippocampal-cerebellar combination or distances from reward were observed, but a significant interaction effect was obtained (distance x combination two-ways repeated measures ANOVA with FDR correction; combination effect, $F_{2,13}$ = 2.33, p = 0.1365; distance effect, $F_{84,1092}$ = 1.043, p = 0.3772; interaction effect, $F_{168,1092}$ = 1.332, p = 0.0053). (E) Same as D for session 20. Significant differences were observed between hippocampal-cerebellar combinations (distance x combination two-ways repeated measures ANOVA with FDR correction; combination effect, $F_{2,13}$ = 6.145, p = 0.0132; distance effect, $F_{84,1092}$ = 1.682, p = 0.0002; interaction effect, $F_{168,1092}$ = 1.271, p = 0.0163). Post-hoc analysis revealed sustained (at least for five consecutive cm) differences between Crus I-HPC and lobule II/III-HPC coherence at distances from −60 to −20 cm from reward (solid green/black line), between lobule VI-HPC and lobule II/III-HPC coherence between −44 and −31 cm from reward (solid red/black line) and also between Crus I-HPC and lobule VI-HPC coherence between −59 and −36 cm from reward (solid green/red line). (F) Top: Averaged LFP power between −60 and −20 cm from reward (session 1). Bottom: Averaged coherence between −60 and −20 cm from reward (session 1). The spurious peak in the 49–51 Hz band generated for the electrical noise has been removed. (G) Same as J for session 20. (H) Averaged theta coherence between −60 and −20 cm from reward between cerebellar recordings and hippocampus during session 1 (left) and session 20 (right, coherence averaged between left and right hemisphere LFPs when both hippocampal recording electrodes were on target; Crus I, n = 4; lobule II/III, n = 6; lobule VI, n = 6). Crus I-HPC coherence was significantly higher than that observed with lobule II/III in the session 20 (*, Kruskal-Wallis with FDR correction, Kruskal-Wallis statistic = 7.989, p = 0.0103; Crus I-HPC vs lobule II/III-HPC, p = 0.0110).

DOI: https://doi.org/10.7554/eLife.41896.016

The following source data and figure supplements are available for figure 4:

**Source data 1.** Cerebello-hippocampal interactions during goal-directed behavior.
DOI: https://doi.org/10.7554/eLife.41896.021

**Figure supplement 1.** Cerebello-hippocampal coherence patterns are conserved across hemispheres during goal-directed behavior.
DOI: https://doi.org/10.7554/eLife.41896.017

**Figure supplement 2.** Distributions and correlations during goal-directed behavior.
DOI: https://doi.org/10.7554/eLife.41896.018

**Figure supplement 3.** Calculation of the imaginary part of coherence during goal-directed behavior.
DOI: https://doi.org/10.7554/eLife.41896.019

**Figure supplement 4.** Cerebello-hippocampal coherence patterns during running or goal-directed movement in a virtual environment.
DOI: https://doi.org/10.7554/eLife.41896.020

p < 0.0001) and the significant increase in their mean speed (*Figure 4B*; mean speed on 1st day = 5.74 ± 0.54 cm/s, mean speed on 7th day = 13.94 ± 1.63 cm/s; repeated measures Friedman test, Friedman statistic = 36.32, p < 0.0001). Thus, we next explored the dynamics of cerebello-hippocampal theta coherence across this learning period. We confirmed the absence of a laterality effect on the power spectra calculated at the beginning (*Figure 4—figure supplement 1A*; Session 1; HPC left, n = 6, median [IQR] 6–12 Hz z-score power = 5.842 [5.683 6.384]; HPC right, n = 7, median [IQR] 6–12 Hz z-score power = 6.261 [4.919 6.587]; Mann-Whitney test, U = 18, p = 0.7308) or end (*Figure 4—figure supplement 1B*; Session 20; HPC left, median [IQR] 6–12 Hz z-score power = 5.850 [5.583 6.002]; HPC right: median [IQR] 6–12 Hz z-score power = 5.898 [5.511 6.110]; Mann-Whitney test, U = 20, p = 0.9452) of training. Consequently, we averaged the spectral power from left and right hemispheres when both were available. Similarly, no differences on coherence between left and right hippocampi and the different cerebellar recordings were observed at the beginning (*Figure 4—figure supplement 1C,E*; Session 1; Crus I-HPC left, n = 3, median [IQR] 6–12 Hz coherence = 0.4789 [0.4557 0.5326]; Crus I-HPC right, n = 3, median [IQR] 6–12 Hz coherence = 0.4802 [0.4687 0.5238]; Mann-Whitney test, U = 4, p > 0.99; lobule II/III-HPC left, n = 4, median [IQR] 6–12 Hz coherence = 0.4574 [0.4513 0.4709]; lobule II/III-HPC right, n = 5, median [IQR] 6–12 Hz coherence = 0.4633 [0.4549 0.4663]; Mann-Whitney test, U = 8, p = 0.7302; lobule VI-

HPC left, n = 6, median [IQR] 6–12 Hz coherence = 0.4764 [0.4653 0.4904]; lobule VI-HPC right, n = 5, median [IQR] 6–12 Hz coherence = 0.4684 [0.4558 0.4857]; Mann-Whitney test, U = 10, p = 0.4286) or end (*Figure 4—figure supplement 1D,F*; Session 20; Crus I-HPC left, median [IQR] 6–12 Hz coherence = 0.5357 [0.4747 0.5574]; Crus I-HPC right, median [IQR] 6–12 Hz coherence = 0.5137 [0.4616 0.5500]; Mann-Whitney test, U = 3, p = 0.7; lobule II/III-HPC left, median [IQR] 6–12 Hz coherence = 0.4596 [0.4466 0.4769]; lobule II/III-HPC right, median [IQR] 6–12 Hz coherence = 0.4595 [0.4422 0.4736]; Mann-Whitney test, U = 10, p = 0.7302; lobule VI-HPC left, median [IQR] 6–12 Hz coherence = 0.4702 [0.4662 0.5124]; lobule VI-HPC right, median [IQR] 6–12 Hz coherence = 0.4665 [0.4610 0.5066]; Mann-Whitney test, U = 12, p = 0.6623) of training so the averaged coherence between cerebellum and both hippocampal hemispheres was computed when possible.

We first examined overall, mean cerebello-hippocampal theta coherence as learning progressed. We observed significant changes over training (*Figure 4C*; Crus I-HPC, n = 4; lobule II/III-HPC, n = 6; lobule VI-HPC, n = 6; day of training x cerebello-hippocampal combination two-way repeated measures ANOVA with FDR correction, day effect $F_{6,84}$ = 3.873, p = 0.0018) and post-hoc analysis revealed that only Crus I-HPC coherence significantly increased when comparing with values observed on the first day of training (p < 0.01 for days 6 and 7). We next examined detailed power spectra and coherence dynamics at the level of individual sessions from first day of training (session 1), when animals exhibited an exploratory behavioral profile, and the last day of training (session 20), when animals performed efficient goal-directed behavior (*Figure 4*). This allowed us to investigate the spatial dynamics of both the cerebellar and hippocampal LFP profiles alongside coherence as mice traversed the linear track to reach the reward.

During session 1, mice approached the reward point with a sustained and low speed (*Figure 4Di*, top), which is in agreement with the exploratory behavioral profile illustrated by the distributed occupancy of their trajectories on the track (*Figure 4A* bottom). The hippocampal spectrogram was dominated by sustained activity in the theta band across the whole track. In contrast, clear activity in this frequency band was not apparent in the cerebellar recordings (*Figure 4Di*) and the LFP power profile was maintained at low levels across the track. This homogeneous pattern was consistent with the unimodal distributions of instantaneous hippocampal and cerebellar theta power (*Figure 4—figure supplement 2A*). Similarly, the coherograms did not reveal clear coherence in any of the cerebello-hippocampal combinations (*Figure 4Dii*). However, we found a significant interaction between the distance from reward and the theta coherence of different cerebello-hippocampal combinations (*Figure 4Diii*; distance from reward point x combination two-way repeated measures ANOVA; combination effect, $F_{2,13}$ = 2.33, p = 0.1365; distance effect, $F_{84,1092}$ = 1.043, p = 0.3772; interaction effect, $F_{168,1092}$ = 1.332, p = 0.0053). Post-hoc, FDR corrected, multiple comparisons revealed that significantly higher theta coherence was present between hippocampus and Crus I compared to lobule II/III at certain positions on the track prior to the reward point location (Crus I-HPC vs lobule II/III-HPC, p < 0.05 from −29 to −21 cm from reward; lobule VI-HPC vs lobule II/III-HPC, p < 0.05 from −25 to −22 cm from reward). As for the homecage recordings, instantaneous theta coherence for all cerebello-hippocampal combinations also followed gaussian, unimodal distributions during session one in the linear track (*Figure 4—figure supplement 2E*).

On the last day of training, during session 20, mice displayed goal-directed behavior on the track. This goal-directed profile was illustrated in the efficient running trajectories (*Figure 4A* bottom) and by the acceleration-plateau-deceleration speed profile observed along the track (*Figure 4Ei* top). As in session 1, the hippocampal spectrogram was dominated by sustained theta activity. On the other hand, cerebellar LFP power profiles were notably different from session one with the appearance of sustained activity in the theta and delta (2–4 Hz) frequency bands, particularly in lobule VI and lobule II/III (*Figure 4Ei* bottom). These differences were not related to transient bouts of theta activity as the instantaneous theta power probability distributions continued showing unimodal profiles (*Figure 4—figure supplement 2C*). Coherogram analysis also revealed changes in session 20, which were mainly reflected in sustained Crus I-HPC theta coherence spanning multiple positions on the track as animals actively approached the reward (*Figure 4Eii*). This pattern was not as apparent in the other cerebello-hippocampal combinations (*Figure 4Eiii*; distance from reward x cerebello-hippocampal combination two-way repeated measures ANOVA, distance from reward effect $F_{84,1092}$ = 1.682, p = 0.0002; combinations effect $F_{2,13}$ = 6.145, p = 0.0132; interaction effect $F_{168,1092}$ = 1.271, p = 0.0163) and post-hoc FDR corrected multiple comparisons revealed significantly higher

sustained theta coherence between hippocampus and Crus I than with lobule II/III from −60 to −20 cm or with lobule VI from - 59 to −36 cm prior to the reward point. Lobule VI-HPC coherence was also higher than with lobule II/III at some positions on the track (−44 to −31 cm from reward) although it was not as sustained and prominent as Crus I-HPC coherence. Importantly, we also reproduced these findings when the imaginary part of coherency was computed, which is robust against contamination by volume conduction (*Figure 4—figure supplement 3*).

In order to better explore the differences between sessions 1 and 20 we then pooled power spectra (*Figure 4F and G* top panels) and coherence (*Figure 4F and G* bottom panels) between −60 and −20 cm from reward point (i.e. the range of distances where differences across recording combinations were most apparent in the coherograms). The appearance of a theta peak in the power spectra from all cerebellar recordings in session 20 compared with session one can be clearly seen (*Figure 4G*) as well as the increase in coherence limited to this frequency band (Session 20: frequency band (theta, beta, low gamma) x cerebello-hippocampal combination two way repeated measures ANOVA, frequency band effect $F_{2,26}$ = 13.42, p < 0.0001; combination effect $F_{2,13}$ = 6.545, p = 0.0108; interaction effect, $F_{4,26}$ = 5.242, p = 0.0031; post-hoc multiple comparisons with FDR correction revealed significant differences between combination only in the theta band). We found that in session 20 theta coherence between hippocampus and Crus I was significantly higher than that obtained with lobule II/III (*Figure 4H* right; Crus I-HPC, n = 4, median [IQR] coherence = 0.525 [0.480–0.549]; lobule II/III-HPC, n = 6, median [IQR] coherence = 0.461 [0.452–0.468]; lobule VI-HPC, n = 6, median [IQR] coherence = 0.470 [0.465–0.503]; Kruskal-Wallis with FDR correction, Kruskal-Wallis statistic = 7.989, p=0.0103; Crus I-HPC vs lobule VI-HPC, corrected p = 0.0110) while no significant difference was found in session 1 (*Figure 4H* left; Crus I-HPC, n = 4, median [IQR] coherence = 0.480 [0.466–0.516]; lobule II/III-HPC, n = 6, median [IQR] coherence = 0.462 [0.457–0.466]; lobule VI-HPC, n = 6, median [IQR] coherence = 0.476 [0.463–0.494]; Kruskal-Wallis with FDR correction, Kruskal-Wallis statistic = 4.779, p = 0.0887).

Given that the speed profiles of mice changed significantly between sessions 1 and 20 on the linear track (*Figure 4B,D,G*) and seemed to mirror the observed modulation in theta power and coherence, we next correlated instantaneous cerebellar theta power (*Figure 4—figure supplement 2B,D*) and instantaneous theta coherence (*Figure 4—figure supplement 2F,H*) with instantaneous speed. In contrast to home-cage recordings, for all recorded cerebellar LFPs, theta power was significantly positively correlated with speed during both session 1 (*Figure 4—figure supplement 2B*; Crus I, median [IQR] Spearman rho = 0.213 [0.209 0.258], bootstrap confidence level = 0.081; lobule II/III, median [IQR] Spearman rho = 0.239 [0.065 0.343], bootstrap confidence level = 0.082; lobule VI, median [IQR] Spearman rho = 0.235 [0.090 0.284], bootstrap confidence level = 0.083) and session 20 (*Figure 4—figure supplement 2D*, Crus I, median [IQR] Spearman rho = 0.235 [0.056 0.465], bootstrap confidence level = 0.081; lobule II/III, median [IQR] Spearman rho = 0.310 [-0.010 0.490], bootstrap confidence level = 0.081; lobule VI, median [IQR] Spearman rho = 0.194 [0.105 0.550], bootstrap confidence level = 0.085). Cerebello-hippocampal theta coherence was not correlated with instantaneous speed during session 1 (*Figure 4—figure supplement 2F*; Crus I-HPC, median [IQR] Spearman rho = 0.024 [-0.033 0.237], bootstrap confidence level = 0.082; lobule II/III-hippocampus, median [IQR] Spearman rho = 0.0278 [-0.105 0.050], bootstrap confidence level = 0.075; lobule VI-HPC, median [IQR] Spearman rho = −0.001 [-0.052 0.045], bootstrap confidence level = −0.090). However, during session 20, lobule II/III-HPC theta coherence was anticorrelated with speed (*Figure 4—figure supplement 2H*; lobule II/III-HPC, median [IQR] Spearman rho = −0.086 [-0.182 0.012], bootstrap confidence level = −0.080) while lobule VI-HPC was weakly but significantly correlated with it (lobule VI-HPC, median [IQR] Spearman rho = 0.091 [-0.140 0.173], bootstrap confidence level = 0.090). Interestingly, the Crus I-HPC recording combination, which presented higher and sustained levels of theta coherence during this task, was not significantly correlated with instantaneous speed (Crus I-HPC, median [IQR] Spearman rho = 0.020 [-0.051 0.094], bootstrap confidence level = 0.083). Together, these findings suggest that the observed Crus I - HPC theta coherence dynamics during goal-directed behavior in the linear track cannot be explained by changes in running speed.

We also observed similar cerebello-hippocampal coherence dynamics in mice navigating for rewards in a virtual reality based linear track (*Figure 4—figure supplement 4A–C*, n = 6). A marked spatial re-organization of cerebello-hippocampal theta coherence was apparent in those mice that showed behavioral modulation across training (as evidenced by increases in the number of rewards

across day of training and speed modulation on a run by run basis during approach to the reward location; *Figure 4—figure supplement 4G–I*). In contrast, mice that failed to show behavioral modulation across training, and displayed rather homogenous speed profiles, did not present such coherence dynamics (*Figure 4—figure supplement 4D–F*).

To examine whether the observed changes in coherence across learning of the linear track were specifically related to performance of the goal-directed task itself, we next conducted pairwise analysis of cerebello-hippocampal theta coherence levels across the following conditions: home-cage prior to any linear track training (HC pre, considered as baseline), the 1st and 20th linear track trials, and home-cage following the end of training in the linear track task (HC post) (see *Figure 5*).

From the three cerebello-hippocampal recording configurations, only Crus I-HPC theta coherence varied significantly across task conditions (*Figure 5*, n = 4, repeated measures Friedman test with FDR correction, Friedman statistic = 11.1, p = 0.0009). At the outset of linear track learning (session 1), HPC-Crus I coherence values did not significantly differ from home-cage (HC pre, median [IQR] = 0.471 [0.461 0.495]; session 1, median [IQR] = 0.480 [0.466 0.516], corrected p = 0.1196). However, during late stage linear track learning, the level of coherence was significantly higher than in home-cage recordings (HC pre, median [IQR] = 0.471 [0.461 0.495]; session 20, median [IQR] = 0.525 [0.480 0.550], corrected p = 0.0021) and when mice were returned to the home-cage environment following completion of linear track training (HC post) the level of HPC-Crus I coherence dropped back to pre-training levels (HC pre, median [IQR] = 0.471 [0.461 0.495]; HC post, median [IQR] = 0.493 [0.471 0.505], corrected p = 0.0580).

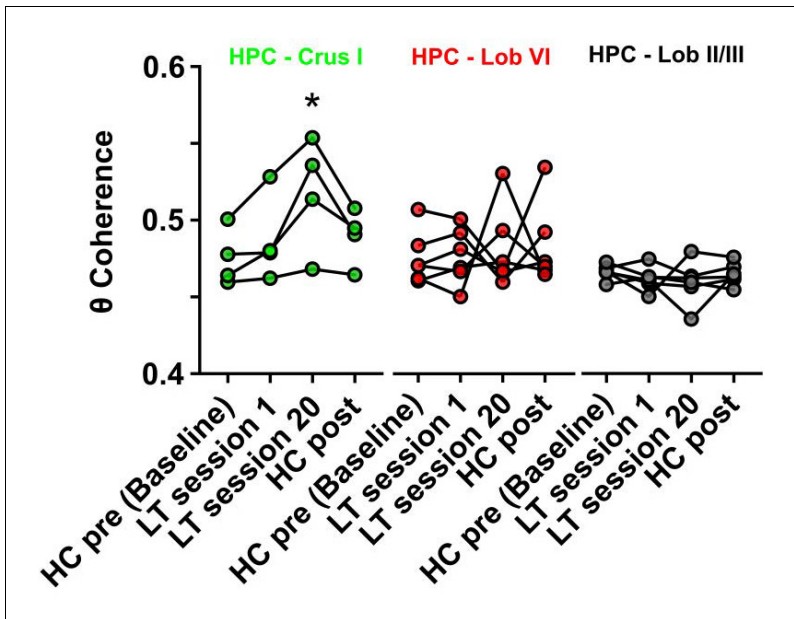

**Figure 5.** Hippocampal-Crus I theta coherence is dynamic. Comparisons of hippocampal-cerebellar theta coherence before and after acquisition of a goal-directed behavior in the linear track task. Levels of coherence during homecage active exploration before first session in the linear track were taken as a baseline. HPC-Crus I theta coherence became significantly different from baseline by session 20 in the linear track task and returned to baseline levels during a second homecage recording immediately following linear track session 20 (n = 4, repeated measures Friedman test with FDR correction, Friedman statistic = 11.1, p = 0.0009; LT session 20 vs baseline, p = 0.0021). No significant differences across conditions were observed for the other hippocampal-cerebellar combinations.

DOI: https://doi.org/10.7554/eLife.41896.022

The following source data is available for figure 5:

**Source data 1.** Hippocampal-Crus I theta coherence is dynamic.

DOI: https://doi.org/10.7554/eLife.41896.023

## Discussion

Taken together, our findings reveal previously undescribed cerebellar inputs to the hippocampus and offer novel physiological insights into a long-range neural network linking disparate brain regions initially assumed to support divergent behavioral functions, namely spatial navigation (hippocampus) and motor control (cerebellum). Projections from topographically restricted regions of cerebellar cortex discretely route through restricted parts of their associated nuclei en-route to the hippocampus through multiple, convergent pathways, involving one or two relays, from the DCN. Interestingly, the possible single-relay pathways we described points toward involvement of the medial septum and the supramammillary nucleus, two structures crucial for theta generation. Congruently, our physiological data suggest that these connected cerebellar regions may dynamically interact with the hippocampus during behavior, via theta (6–12 Hz) LFP coherence. Our findings thus offer an anatomical and physiological framework for cerebello-hippocampal interactions that could support cerebellar contributions to hippocampal processes (*Burguière et al., 2005*), including spatial map maintenance (*Rochefort et al., 2011*; *Rondi-Reig et al., 2014*; *Lefort et al., 2019*).

Whilst previous studies provide compelling physiological evidence of cerebellar influences on the hippocampus (*Cooke and Snider, 1955*; *Iwata and Snider, 1959*; *Babb et al., 1974*; *Snider and Maiti, 1975*; *Krook-Magnuson et al., 2014*; *Choe et al., 2018*), they do not provide the spatial resolution afforded by neuroanatomical tracing. Indeed, to the best of our knowledge, anatomical tracing studies have failed to report a mono-synaptic ascending cerebello-hippocampal projection. This is consistent with our rabies virus tracing study, in which incubation periods of 48–58 hr were required before cell labeling was seen in the cerebellar nuclei. Such a timescale is indicative of a multi-synaptic pathway (*Kelly and Strick, 2000*; *Ugolini, 2010*; *Suzuki et al., 2012*; *Jwair et al., 2017*). Single relay pathways can be envisioned through the septum, the hypothalamus (potentially including the supramammillary nucleus (SUM)) and the raphe nucleus. Other pathways including two relays through either the lateral and medial entorhinal cortex and/or the perirhinal cortex are also possible. Interestingly, among the different regions labeled at 48 hr post-infection, several midbrain and pontine regions such as the PAG, the nucleus incertus and the LtDG are known to receive direct projections from the DCN and could therefore represent putative first-order relays between cerebellum and hippocampus.

Our anatomical results highlight three main inputs to the hippocampus emanating from the cerebellum. The first input we reveal originates from the vestibulo-cerebellum, specifically from the dorsal and ventral paraflocculus, which is likely routed via the vestibular and dentate nuclei (*Voogd and Barmack, 2006*). This anatomical connection between the vestibulo-cerebellum and the hippocampus reinforces the already well-described influence of the vestibular system on hippocampal-dependent functions (*Stackman et al., 2002*; *Goddard et al., 2008*; *Zheng et al., 2009*).

In addition to the classically described vestibular pathway, our data reveal that the central cerebellum also provides inputs to the hippocampus from vermal lobule VI, routed through caudal fastigial nucleus, and from Crus I, routed through the dentate. Using a combination of RABV expression and zebrin II staining, we identified three specific cerebellar modules involved in these inputs: (1) the A module in lobule VI, (2) the hemispheric Crus I D2 module and (3) the Crus I paravermal C2 module. Of the latter two modules, C2 is likely less prominently anatomically connected with the hippocampus since the number of RABV+ cells in the nucleus interpositus posterior, its output nucleus (*Apps and Hawkes, 2009*), was minor compared with the other cerebellar nuclei. The convergence of inputs from disparate cerebellar zones (flocculo-nodular and central zones) and modules from vermal (A), paravermal (C2) and hemispheric (D2) regions in to the hippocampus suggest that its optimal function requires the integration of multiple aspects of sensory-motor processing carried out at these distinct cerebellar locations.

According to *Voogd and Barmack (2006)*, based upon evidence from a wide range of species, the oculomotor cerebellum can be most broadly described to include lobule V, VI and VII. Given the highly conserved structure-function relationships of the vermal cerebellum (*Sillitoe et al., 2005*), it seem likely that the mouse A module could also be considered part of the oculomotor cerebellum. In rat, it receives climbing fibers from the caudal medial accessory olive, and sends mainly ascending projections through the caudal portion of fastigial nucleus (*Apps, 1990*; *Apps and Hawkes, 2009*). The oculomotor vermis receives multiple sensory inputs which include visual, proprioceptive, vibrissae, vestibular and auditory inputs conveyed by both climbing and mossy fibers (*Voogd and*

*Barmack, 2006*). The D2 module receives its climbing fiber input from the dorsal cap of the principal olive and projects out of the cerebellum through the rostromedial dentate nucleus (*Herrero et al., 2006*). It receives mossy fiber inputs carrying somatosensory, motor (*Mihailoff et al., 1981*), and visual (*Edge et al., 2003*) information; along with inputs from the prefrontal cortex (*Kelly and Strick, 2003*). Climbing fiber inputs to this module relay information from the parvocellular red nucleus, which receives projections from premotor, motor, supplementary motor and posterior parietal areas. The majority of these cortical areas also receive projections from the D2 module after a thalamic relay in the ventro-lateral nucleus (*Kelly and Strick, 2003*; *Glickstein et al., 2011*).

Complementary to these anatomical results, our electrophysiological findings reveal coherent activity between the hippocampus and those cerebellar lobules that are anatomically connected with it (lobule VI and Crus I). This synchronization was restricted to the 6–12 Hz frequency range in the awake, behaving animal and showed dynamic profiles that were lobule dependent. Oscillations can align neuronal activity within and across brain regions, suggesting a facilitation of cross-structure interactions (e.g. *Singer, 1999*; *Fries, 2005*). Cerebellar circuits support oscillations across a range of frequencies (for review see *De Zeeuw et al., 2008*; *Cheron et al., 2016*). Of particular relevance to the current study are reports of oscillations within the theta frequency (~4–12 Hz), which have been described in the cerebellar input layers at the Golgi (*Dugué et al., 2009*) and granule cell (*Hartmann and Bower, 1998*; *D'Angelo et al., 2001*) level, and also in the cerebellar output nuclei (*Wang et al., 2014*; *Watson et al., 2014*).

Neuronal coherence has been described across the cerebro-cerebellar system at a variety of low frequencies (*O'Connor et al., 2002*; *Courtemanche and Lamarre, 2005*; *Soteropoulos and Baker, 2006*; *Rowland et al., 2010*; *Frederick et al., 2014*; *Watson et al., 2014*; *Chen et al., 2016*) and oscillations within the theta range are thought to support inter-region communication across a wide variety of brain regions (*Colgin, 2013*). Our finding that cerebello-hippocampal coherence is limited to the 6–12 Hz bandwidth is in keeping with previous studies on cerebro-cerebellar communication in which neuronal synchronization has been observed between the cerebellum and prefrontal cortex (*Watson et al., 2014*; *Chen et al., 2016*), primary motor cortex (*Soteropoulos and Baker, 2006*; *Rowland et al., 2010*), supplementary motor area (*Rowland et al., 2010*) and sensory cortex (*Rowland et al., 2010*). Furthermore, LFPs recorded in the hippocampus and cerebellar cortex are synchronized within the theta bandwidth during trace eye-blink conditioning in rabbits (*Hoffmann and Berry, 2009*; *Wikgren et al., 2010*). Human brain imaging studies have also described co-activation of blood oxygen level dependent signals in both cerebellar and hippocampal regions during navigation (*Iglói et al., 2015*) and spatio-temporal prediction tasks (*Onuki et al., 2015*), thus highlighting putative neuronal interactions between the two structures. Regarding studies in mice, a recent study has demonstrated the existence of statistically significant co-activation of the dorsal hippocampus and cerebellar lobules IV-V, lobule VI and Crus I after the acquisition of a sequence-based navigation task (*Babayan et al., 2017*).

Multiple lines of evidence suggest that the coherence described here is unlikely to have resulted from volume conduction: 1) Rather than using a common reference electrode, our recordings were bipolar, with each recording electrode being locally and independently referenced (*Kajikawa and Schroeder, 2011*). 2) If volume conduction of theta oscillations was emanating from a hippocampal source then it could be assumed that cerebellar regions in closer proximity to the hippocampus would show higher levels of coherence (*Figure 3—figure supplement 2*). However, we found that coherence values were not related to the relative distance between the hippocampus and cerebellar recording site. 3) By recording simultaneously from hippocampus and multiple cerebellar regions, we have been able to demonstrate that the observed coherence is non-homogenous among the different cerebellar lobules in contrast to what one would expect if theta was volume conducted from a common location. 4) We calculated the imaginary part of coherency, which is not influenced by volume conduction (*Nolte et al., 2004*), in the linear track paradigm and found results that were remarkably similar to those obtained using standard coherence analysis. 5) We showed that Purkinje cell spikes in lobule VI of the cerebellum can phase lock to hippocampal theta oscillations.

Importantly, we have shown for the first time that theta rhythms in the hippocampus preferentially synchronize with those in discrete regions of the cerebellum and that the degree of this coupling changes depending upon the behavioral context. Lobule VI-HPC coherence was dominant during active movement in the home-cage and remained stable during learning of the real world linear track task. On the other hand, Crus I - HPC coherence was highly dynamic, showing a significant

increase over learning of the real world linear track task and becoming dominant after the acquisition of a goal-directed behavior. Furthermore, we reveal that high Crus I-HPC coherence was sustained across the linear track when the animal performed goal-directed behavior but not during the early training, exploratoration phase. Such sustained coherence may be related to the ability of the cerebellum, and particular Crus I, to link internal and external sensory context with specific action toward the goal. In line with this hypothesis, a recent study has suggested that Purkinje cells in the Crus I region and neurons in its output, the dentate nucleus, exhibit firing rate modulation in anticipation of expected reward in a VR task (*Chabrol et al., 2018*). Interestingly, they show that the activity of Purkinje cells in lateral Crus I was modulated by running and/or visual flow speed. This is consistent with our examples showing that Crus I-HPC coherence remained present when animals performed goal-modulated behavior in VR, although multiple streams of sensory input, such as vestibular, whisker and olfactory, become irrelevant and even confounding in the head-restricted virtual environment task.

We next consider our results within the well characterized, modular understanding of cerebellar function. Within lobule VI, the A module receives multi-modal sensory information, mainly arising from collicular and vestibular centers (*Voogd and Barmack, 2006*). The superior colliculus plays a role in visual processing and generation of orienting behaviors (*Basso and May, 2017*), which might be relevant for the establishment and maintenance of the hippocampal spatial map, and thus may be required constantly during active movement, independent of the specific behavioral task. The persistent and similar levels of lobule VI-HPC coherence during active movement in the homecage and linear track task, in both real world and virtual reality environment tasks is in agreement with such a hypothesis.

In monkeys and humans, Crus I is anatomically and functionally associated with prefrontal cortex (*Kelly and Strick, 2003*; *Iglói et al., 2015*). In mouse Crus I, the D2 module receives convergent sensory and motor information (*Proville et al., 2014*). Furthermore, this module has been found to contain internal models, a neural representation of one's body and the external world based on memory of previous experiences, that are used for visuo-motor coordination (*Cerminara et al., 2009*). Similarly, the C2 module has been found to also participate in visuo-motor processing related to limb coordination during goal-directed reaching (*Cerminara and Apps, 2011*). Both modules might be particularly important during the acquisition of a goal-directed behavior such as our real-world linear track task in which animals needed to reach non-cued reward zones. Our finding that Crus I-HPC coherence increases during task learning fits with this hypothesis.

In summary, our results suggest the existence of anatomically discrete hippocampal-cerebellar network interactions with a prominent involvement of Crus I during goal-directed behavior. Both anatomical and electrophysiological data point toward involvement of the theta generating pathway in cerebellum-hippocampus interactions. The topographical dynamic weighting of these interactions may be tailored to the prevailing sensory context and behavioral demands.

## Materials and methods

Anatomical tracing studies were performed under protocol N°00895.01, in agreement with the Ministère de l'Enseignement Supérieur et de la Recherche. RABV injections were performed by vaccinated personnel in a biosafety containment level two laboratory.

All behavioral experiments were performed in accordance with the official European guidelines for the care and use of laboratory animals (86/609/EEC) and in accordance with the Policies of the French Committee of Ethics (Decrees n° 87–848 and n° 2001–464). The animal housing facility of the laboratory where experiments were made is fully accredited by the French Direction of Veterinary Services (B-75-05-24, 18 May 2010). Surgeries and experiments were authorized by the French Direction of Veterinary Services (authorization number: 75–752).

A total of 44 adult, male mice were used for this study. Seventeen adult male C57BL6-J mice were used for the anatomical tracing study, (Charles River, France) and 21 for the electrophysiology study (Janvier, France). Six adult male CD-L7ChR2 mice were used for the dual hippocampal LFP and cerebellar unit-recording study (in-house colony derived from Jackson labs stock, USA).

Mice received food and water *ad libitum*, were housed individually (08: 00–20: 00 light cycle) following surgery and given a minimum of 5 days post-surgery recovery before experiments commenced.

## Anatomy

### Rabies virus injections

All the RABV (the French subtype of Challenge Virus Standard; CVS-N2C) inoculations were performed in the Plasticity and Physio-Pathology of Rhythmic Motor Networks (P3M) laboratory, Timone Neuroscience Institute, Marseille, France. Mice (n = 17) were injected intraperitoneally with an anesthetic mixture of ketamine (65 mg/kg; Imalgene, France) and xylazine (12 mg/kg; Rompun, Bayer) to achieve surgical levels of anesthesia, as evidenced by the absence of limb withdrawal and corneal reflexes and lack of whisking and were then placed in a stereotaxic frame (David Kopf Instruments, USA). The scalp was then incised, the skull exposed and a craniotomy drilled above the hippocampus.

Mice were injected with 200 nL of a mixture of one part 1% CTb Alexa Fluor 488 Conjugate (Invitrogen, distributed by Life Technologies, Saint Aubain, France) and four parts RABV in the left hippocampus (AP −2.0, ML +2.0, DV 1.97; *Figure 1—figure supplement 1 and* ) using a Hamilton needle with internal diameter of ~200 µm (Hamilton, USA). Injections (200 nL/min) were performed using a pipette connected to a 10 µL Hamilton syringe mounted on a microdrive pump. Following infusion, the pipette was left in place for 5 min. The incision was then sutured and the animals allowed to recover in their individual home cage for either 30 hr (n = 4); 48 hr (n = 3), 58 hr (n = 5) or 66 hr (n = 5). All animals were carefully monitored during the survival period and, in line with previous studies using these survival times, were found to be asymptomatic (*Ugolini, 2010*).

### Tissue preparation

At the end of the survival time, mice were deeply anesthetized with sodium pentobarbitone (100 mg/kg, intraperitoneal) then transcardially perfused with 0.9% saline solution (15 mL/min) followed by 75 mL of 0.1M phosphate buffer (PB) containing 4% paraformaldehyde (PFA; pH = 7.4). The brain was then removed, post fixed for 2–3 days in 4% PFA and then stored at 4°C in 0.1 M PB with 0.02% sodium azide. Extracted brains were then embedded in 3% agarose before being coronally sectioned (40 µm) on a vibratome. Serial sections were collected and divided in 4 vials containing 0.1 M PB so consecutive slices in each vial were spaced by 160 µm.

### Injection site visualization

Sections from vial one were used to visualize the injection site by the presence of CTb. In most of the cases, the injected CTb was fluorescent and sections were directly mounted with Dapi Fluoromount G (SouthernBiotech, Alabama, USA). In the other cases (S4-5, S11-13 and S17-18), the sections were first rinsed with PB 0.1 M and then permeated with PB 0.1 M and 0.3% Triton X-100. They were then incubated overnight in a choleragenoid antibody raised in goat (goat anti-CTb, lot no. 703, List Biological Laboratories, USA) diluted 1: 2000 in a blocking solution (PB 0.1 M, 5% BSA). Subsequently, the sections were rinsed in PB 0.1 M and incubated for 4 hr at room temperature with donkey anti-goat secondary antibody (1: 1000 in the blocking solution; Alexa Fluor 555, Invitrogen, distributed by ThermoFisher Scientific, Massachusetts, USA). Finally, the sections were mounted with Dapi Fluoromount G.

The injection site was then visualized using a fluorescence microscope equipped with a fluorescein isothiocyanate filter (Axio Zoom V16, Carl Zeiss, France).

### Rabies virus labeled cell quantification

Sections from vial two were used for quantification and 3 D reconstruction of the RABV labeled cells. Sections mounted on gelatin-coated SuperFrost Plus slides (Menzel-Glaser, Braunschweig, Germany) were first rinsed with PB 0.1 M and pre-treated with 3% $H_2O_2$ for 30 min in the blocking reaction against endogenous peroxidase. Following pretreatment, the sections were incubated overnight at room temperature with an anti-rabies phosphoprotein mouse monoclonal antibody (*Raux et al., 1997*) diluted at 1: 10000 in a blocking solution (PB 0.1 M, 0.1% BSA, goat serum 2% and 0.2% Triton X-100). The next sections were rinsed in PB 0.1 M and incubated 2 hr with a biotinylated affinity-purified goat anti-mouse IgG (1: 2000 in blocking solution; Santa-Cruz, Heidelberg, Germany). Then, they were also incubated using an avidin-biotin complex method (Vectastain Elite ABC-Peroxidase kit R.T.U. Universal, Vector Laboratories, Burlingame, CA, USA) to enhance sensitivity. For visualization, the sections were incubated in a 3,3'-diaminobenzidine-tetrahydrochloride (DAB) solution

(0.05% DAB and 0.015% $H_2O_2$ in PB 0.1 M). Finally, they were counterstained with cresyl and cover-slipped.

Quantitative analyses of rabies-positive cells were performed using a computerized image processing system (Mercator, Exploranova, France) coupled to an optical microscope. The quantification of rabies-positive cells was carried out at 10x magnification. Structures were defined according to a standard atlas (*Franklin and Paxinos, 2007*). Immunoreactive neurons were counted bilaterally. Representative images were obtained using an Axio Zoom V16 microscope (Carl Zeiss, France).

### 3-D reconstruction
A Nikon Eclipse E800 microscope equipped with a digital color camera (Optronics, USA) was used to visualize mounted cerebellar sections under brightfield illumination. The contour of every fourth section was then manually drawn using Microfire software (Neurolucida, MBF Bioscience, USA) and cell counts were performed. The sections were then aligned and stacked (160 μm spacing).

### Rabies virus-zebrin II double immuno-staining
For case S18, sections from vial three were mounted on gelatin-coated SuperFrost Plus slides (Menzel-Glaser, Braunschweig, Germany), rinsed with PB 0.1 M and then permeated and blocked in a solution of PB 0.1 M, 0.2% Triton X-100 and bovine serum 2.5% for 30 min. Then they were incubated for 48 hr at 4°C in a mix of rabbit polyclonal anti-Aldolase C primary antibody (a kind gift from Izumi Sugihara (*Sugihara and Shinoda, 2004*); No. 69075; 1:500000) and the mouse anti-rabies antibody used for the single RABV staining (1:5000) in a blocking solution (PB 0.1 M, 0.1% Triton X-100 and bovine serum 1%). Subsequently, the sections were first rinsed with PB 0.1% and then incubated in a mix of Rhodamine Red-XGoat anti-rabbit IgG (1: 5000; ref 111-295-144, Jackson Immuno Research) and donkey anti-mouse secondary antibody (1: 5000; Alexa Fluor 647, Invitrogen distributed by ThermoFisher Scientific, Massachusetts, USA) in blocking solution. Finally, the sections were mounted with Dapi Fluoromount G.

Images were obtained using an Axiozoom v16 microscope (Carl Zeiss, France) then cerebellar contours and labeled neurons were manually drawn for reconstruction of zebrin bands,cerebellar modules and location of the RABV+ cells.

## Electrophysiology procedures
### Preparation 1: Dual hippocampal – cerebellar LFP recording
#### Subjects and surgical protocols
Bipolar LFP recording electrodes (interpolar distance of ~0.5 mm; 140 μm diameter Teflon coated stainless-steel, A-M system, USA) were stereotaxically targeted to hippocampus (AP −2.2, ML +2.0, DV 1.0), lobule VI (AP −6.72, ML 0.0, DV 0.1), lobule II/III (AP −5.52, ML 0.0, DV 1.8) and Crus I (AP −6.24, ML 2.5, DV 0.1) of 21 C57BL6-J mice. Pairs of flexible stainless-steel wires were used to also record neck EMG (Cooner wire, USA).

In 15 C57BL6-J mice, bipolar stimulation electrodes (140-μm-diameter stainless steel; A-M system, USA) were also implanted at the left medial forebrain bundle [MFB; to serve as a reward signal; AP −1.4, ML +1.2, DV +4.8 (*Carlezon and Chartoff, 2007*; *de Lavilléon et al., 2015*). All electrode assemblies were fixed to the skull using a combination of UV activated cement (SpeedCem, Henry Shein, UK), SuperBond (SunMedical, Japan) and dental cement (Simplex Rapid, Kemdent, UK). Four miniature screws (Antrin, USA) were also attached to the skull for additional support and to serve as recording ground.

In six mice, a lightweight metal head fixation device (0.1 g) was also affixed to the implant. The total implant weight did not exceed 2.5 g (including head fixation post and cement).

#### Recording
Electrodes were attached to an electronic interface board (EIB 18, Neuralynx, USA) during surgery. Differential recordings were made via a unity-gain headstage preamplifier (HS-18; Neuralynx, USA) and Digital Lynx SX acquisition system (Neuralynx, USA). LFP and EMG Signals were bandpass-filtered between 0.1 and 600 Hz and sampled at 1 kHz. Mouse position was tracked at 30 Hz using video tracker software and infra-red LEDs attached to the headstage (Neuralynx, USA).

## Medial forebrain bundle (MFB) stimulation

Intracranial rewarding stimulation consisted of a 140 Hz stimulation train lasting 100 ms delivered through the headstage to the implanted electrodes (SD9k, Grass Technologies, USA). The optimal voltage for intracranial MFB was determined for each mouse with a nose-poke task prior to training (range, 1–6 V [*de Lavilléon et al., 2015*]).

# Preparation 2: Simultaneous cerebellar single unit and hippocampal LFP recording

## Subjects and surgical protocols

General surgery procedures were similar to those used for preparation 1 (dual hippocampal – cerebellar LFP recording experiments). CD-L7ChR2 mice (selectively expressing ChR2 in Purkinje cells; n = 6 mice) were implanted with bipolar LFP recording electrodes in the left hippocampus (AP −2.2, ML +2.0, DV 1) and a lightweight (<0.1 g) recording chamber was constructed over the cerebellum. A silicon elastomer (QuickSil, World Precision Instruments, USA) was used to seal the chamber following surgery and between recording sessions. A stainless steel post (0.1 g) was also affixed to the skull and used for head fixation. To measure movements, flexible stainless steel EMG electrodes were implanted in the front left forelimb.

## Recording

Animals were positioned in a custom-built head fixation device and placed on a floating Styrofoam ball (see virtual reality Materials and methods for further details). Hippocampal and EMG electrodes were connected to an electronic interface board (EIB 18, Neuralynx, USA). Filter and recording settings were the same as in preparation 1. For cerebellar single unit recordings, the silicon elastomer was removed from the recording chamber allowing the exposed cerebellar cortex to be viewed under a microscope (Leica, USA). Quartz based electrodes or tetrodes (impedance 1 mOhm at 1 Khz; Thomas Recording, Germany) were inserted in to the cerebellar cortex using a custom manipulator mounted on a stereotaxic frame (Kopf, USA). An optical fiber (600 µm diameter, Prizmatix, Israel) was mounted on a separate stereotaxic manipulator and positioned inside the chamber just above the cerebellar cortical surface within close proximity to the recording electrode. A low impedance silver ball reference electrode was also positioned within the chamber and a skull screw served as ground. A hydraulic micromanipulator was used to lower the optetrodes through the cerebellar cortical layers (Narishige, Japan).

Once units were identified, blue light pulses were used to photo-identify putative Purkinje cells (50 mW/mm$^2$, ~460 nm, 100 ms delivered using a commercial LED driver, Prizmatix, Israel. *Chaumont et al., 2013*). Cerebellar signals were recorded using an EIB 18 and headstage (HS-18, Neuralynx, USA) connected to the electrodes via a custom made adapter. Unit signals were band-pass filtered between 0.3 and 9 kHz while sampling was set at 32 kHz.

## Histology

After completion of all the experiments, mice were deeply anesthetized with ketamine/xylazine solution (150 mg/kg) and electrolytic lesions created by passing a positive current through the electrodes (30µA, 10 s). With the electrodes left in situ, the animals were perfused transcardially with saline followed by paraformaldehyde (4%).

Brains were extracted and post-fixed in paraformaldehyde (4%; 24 hr) then embedded in agarose (24 hr). A freezing vibratome was used to cut 50 µm thick sagittal cerebellar and coronal hippocampal sections. The sections were mounted on gelatinized slides and stained with cresyl violet. Recording locations were identified by localized lesions in the cerebellum and hippocampus and plotted on standard maps with reference to a stereotaxic atlas (*Franklin and Paxinos, 2007*).

# Behavioral procedures

## Familiar environment

All recordings were made in the animal's home-cage (30 cm x 10 cm x 10 cm plastic box), with the lid removed and lasted a maximum of 4 hr. Recordings were made during the day between the hours of 10 am and 6 pm.

### Linear track – real world

The linear track was made in-house from 100 cm x 4 cm x 0.5 cm of black plastic positioned 20 cm above the surface of the experimental table. The behavioral assembly was located in a separate room from the experimenter and was surrounded on four sides by black curtains. Three salient visual cues were placed at fixed locations along the edge of the track (10 cm from the edge). Mice were trained to run in a sequential manner from one end of the track to the other in order to receive a reward, which consisted of an electrical stimulation of the MFB. The reward stimulation was delivered automatically when the mice reached a 5 cm wide goal-zone, which was located 10 cm from the end of the track. Timing of the reward signal was logged on the electrophysiological recordings via TTL signals. Sessions lasted 12 min and were repeated three times per day with an inter-session time of 5 min over 7 days. Between sessions, the track was cleaned with 20% ethanol.

### Linear track – virtual reality environment

A commercially available virtual-reality environment was used (Jet Ball, Phenosys, Germany), utilizing an air cushioned Styrofoam ball (200 mm), which served as a spherical treadmill for head restrained mice (*Lasztóczi and Klausberger, 2016*) (*Figure 4—figure supplement 2*). The floating ball assembly was positioned 20 cm from a series of six octagonally arranged TFT surround monitors (19 inch) such that the head restrained mice had an unobstructed view of the visual scene. Rotation of the Styrofoam ball was detected by an optical sensor (sampling frequency 5700 dots per inch at 1 kHz). The vertical axis signals were interpreted by the VR software as the forward and backward movement of the virtual position of the animal. Position within the VR was then translated to a voltage signal (zero to five volts, with five volts indicating the end of the track), and sent to the Digital Lynx SX (Neuralynx, USA) electrophysiology system via a DACQ interface (DACQ 6501, National Instruments, USA). The start of the VR display was logged on the electrophysiology recordings via a TTL signal. To provide a reward signal, when the mice reached a given location within the VR (10 cm from the end of the track) a TTL marker was sent to both the electrophysiological recording system (to provide a timestamp-marker of the event) and an electrical stimulus generator linked to the HS-18 headstage (in the same manner as for real world linear track experiments).

The virtual scene consisted of a 1 m long track with gray walls and included three salient visual cues. After 3 × 12 mins sessions of habituation to the head fixation on the floating-ball assembly, mice were trained to run on the linear track in 12 min sessions, three times per day with an inter-session interval of approximately 5 mins during 7 days. The number of rewards received by the animal was logged in the electrophysiology software (Cheetah 5.6.3, Neuralynx, USA).

## Behavioral and electrophysiological analysis

All data were processed in Matlab (Mathsworks, USA), Spike 2 (Cambridge Electronic Design, UK) and Prism (Graphpad, USA).

### Behavior

In all conditions, behavioral data were analyzed using custom-made Matlab scripts. Instantaneous speed (or virtual speed) was derived from video tracking data (or virtual X and Y coordinates recorded as voltage signals in Neuralynx Cheetah software) and downsampled to 10 Hz for consistency with spectrogram and coherogram data (see below). Only epochs of active movement, defined by a sustained speed above 3 cm/s for a minimum of 4 s, were selected for further analysis.

In the home-cage environment, there was one case in which video tracking was not available and the epochs of active movement were thus defined by an amplitude threshold in EMG signal and the presence of a high theta/delta ratio in the hippocampal recordings.

For linear track and virtual reality-based experiments, active movement periods (instantaneous speed >3 cm/s) were analyzed from each 12 min session (three sessions per day) for overall calculations. Each session was then subdivided in to trials (runs), that is, each time the animal traversed the track from one reward site to the other, and measurements of speed, power and coherence were averaged by distance from reward location in 1 cm bins normalized by the occupancy (time spent in each bin).

## Electrophysiology

LFP data was z-scored, notch filtered (filter centered to 50 Hz to remove electrical line noise) and detrended (using local-linear regression with a moving window of 1 s in 0.5 s steps to remove the DC component) prior to subsequent analysis. Multi-taper Fourier analyses (Chronux toolbox [*Bokil et al., 2010*]) were used to calculate power and coherence of the LFP data. We used a 1 s sliding window in 0.1 s step and four tapers for all analysis. Time points with large-amplitude, low frequency artifacts, identified by threshold crossings of the mean z-score power in 0.5–5 Hz band, were removed from further analysis. Similarly, peri-MFB electrical stimulation times (±0.5 s) in linear track and virtual environment conditions were also excluded from analysis.

For overall power spectra and coherence calculations, the means of the spectrograms and coherograms were respectively computed. Spectral power between 0.1 and 500 Hz was z-scored to homogenize values and reduce the impact of inter-animal and inter-region global variations. Frequency axes in both spectral power and coherence plots were presented in logarithmic scale to facilitate visualization across a large frequency band (1–300 Hz). Frequencies between 48–52 Hz were removed from coherence plots due to the presence of a spurious peak related to the notch filtering. Data duration of recordings made in the home-cage environment, varied across mice (range, 12 to 132 min). Therefore, to reduce the impact of data length on subsequent analyses and also to match with subsequent linear track experiments (duration of 12 min), for each mouse we concatenated the LFP in to 12 min blocks. When multiple 12 min blocks were available (number of data blocks ranged from 1 to 11), we calculated the average coherence across all blocks.

## Combined recordings of cerebellar units and hippocampal LFP

Cerebellar cell recordings were sorted using Spike2 software (CED, UK) where single units were verified with principal component analysis. We did not separate simple and complex spikes when calculating firing rate. For each cell, the firing rate was normalized against a 1 s pre photo-illumination period. Firing rate was then computed in 10 ms bins. A change of 1.96z was used to classify a significant change in firing rate during photo-illumination (*Chaumont et al., 2013*). For phase locking analysis, hippocampal LFP was bandpass filtered from 6 to 12 Hz. Circular statistics were used to quantify phase distribution (e.g. *Jones and Wilson, 2005*) of each cell and to determine significant phase-locking at p < 0.05.

## Statistical analysis

Statistical analyses were conducted using Matlab Statistical Toolbox and Prism (Graphpad, USA). Normality was assessed using a Shapiro-Wilk test. Parametric and non-parametric tests were then used accordingly.

For bootstrap calculations on the correlation of theta power and theta coherence with speed, we computed the Spearman's correlation between these variables and a randomly shuffled rearrangement of instantaneous speed values. We repeated this 1000 times in order to obtain the cumulative probability distribution of the random correlation values. The confident limits for $\alpha = 0.05$ were obtained by finding the correlation values at probabilities of 0.025 and 0.975.

## Acknowledgements

This work was supported by the Fondation pour la Recherche Médicale DEQ20160334907-France, by the National Agency for Research ANR-17-CE16-0019-03 (LRR), by the CNRS and Aix-Marseille Université through UMR 7289 (PC). This work also received support under the program Investissements d'Avenir launched by the French Government and implemented by the ANR, with the references, PER-SU (LRR) and ANR-10-LABX-BioPsy (LRR). The group of LRR is member of the Labex BioPsy and ENP Foundation. Labex are supported by French State funds managed by the ANR within the Investissements d'Avenir programme under reference ANR-11-IDEX-0004–02. We thank Roxanna Ureta for help with histology, Lilith Sommer for help with behavioral experiments, Gregory Sedes and Nadine Francis for help developing analysis codes, and Richard Apps for his insightful comments. We are grateful to Richard Hawkes and Izumi Sugihara for generously providing the aldolase C antibody. Finally, we thank all members of the CEZAME team for helpful discussions of the experiments and manuscript.

## Additional information

### Funding

| Funder | Grant reference number | Author |
|---|---|---|
| Fondation pour la Recherche Médicale | DEQ20160334907 | Laure Rondi-Reig |
| Agence Nationale de la Recherche | ANR-17-CE16-0019-03 | Laure Rondi-Reig |
| Centre National de la Recherche Scientifique | | Patrice Coulon |
| Aix-Marseille Université | | Patrice Coulon |
| Université Pierre et Marie Curie | ANR-10-LABX-BioPsy | Laure Rondi-Reig |
| Université Pierre et Marie Curie | ANR-11-IDEX-0004-02 | Laure Rondi-Reig |

The funders had no role in study design, data collection and interpretation, or the decision to submit the work for publication.

### Author contributions

Thomas Charles Watson, Formal analysis, Supervision, Validation, Investigation, Methodology, Writing—original draft; Pauline Obiang, Validation, Investigation, Methodology, Writing—review and editing; Arturo Torres-Herraez, Software, Formal analysis, Validation, Investigation, Visualization, Methodology, Writing—original draft; Aurélie Watilliaux, Formal analysis, Validation, Investigation, Methodology, Writing—review and editing; Patrice Coulon, Resources, Supervision, Investigation, Methodology, Writing—review and editing; Christelle Rochefort, Validation, Methodology, Writing—review and editing; Laure Rondi-Reig, Conceptualization, Resources, Supervision, Funding acquisition, Methodology, Writing—original draft, Project administration

### Author ORCIDs

Thomas Charles Watson (iD) https://orcid.org/0000-0002-5146-6981
Arturo Torres-Herraez (iD) https://orcid.org/0000-0001-6333-5623
Patrice Coulon (iD) https://orcid.org/0000-0003-2996-405X
Laure Rondi-Reig (iD) https://orcid.org/0000-0003-1006-0501

### Ethics

Animal experimentation: This study was performed in strict accordance with the recommendations in the Guide for the Care and Use of Laboratory Animals of the Ministère de l'Enseignement Supérieur et de la Recherche. All of the animals were handled according to approved institutional animal care agreement B75 0510. The protocol was approved by the Committee on the Ethics of Animal Experiments (expérimentation animale n°5) protocols #00895.02 and APAFIS#4315-2016042708195884v1. All surgery was performed under isoflurane gas anesthesia, and every effort was made to minimize suffering.

### Decision letter and Author response

Decision letter https://doi.org/10.7554/eLife.41896.026
Author response https://doi.org/10.7554/eLife.41896.027

## Additional files

### Supplementary files

• Transparent reporting form
DOI: https://doi.org/10.7554/eLife.41896.024

#### Data availability

The individual values are plotted in all the graphs and the mean and SEM are also plotted or indicated at the legends. The Matlab codes used for the spectral analysis are part of a fully available, well documented toolbox (Chronux toolbox) and referenced in the text. The parameters used for these codes is indicated in the materials and methods section.

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
