## [Decision Letter]

Thank you for submitting your article "Anatomical and physiological foundations of cerebello-hippocampal interactions" for consideration by *eLife*. Your article has been reviewed by three peer reviewers, and the evaluation has been overseen by Indira Raman as the Reviewing Editor and Richard Ivry as the Senior Editor. The following individual involved in the review of your submission has agreed to reveal his identity: Peter Strick (Reviewer #3).

The reviewers have discussed the reviews with one another and the Reviewing Editor has drafted this decision to help you prepare a revised submission.

Summary:

This manuscript examines the communication between the cerebellum and the hippocampus by tracing anatomical pathways connecting the two regions and examining the coherence of theta oscillations between the (dorsal) hippocampus and the cerebellum during spatial navigation in mice.

Essential revisions:

The three reviewers all found aspects of the work to be of interest. They all, however, had fairly extensive critiques, including comments on the limitations of the correlative data and the ambiguity of the link between cerebellar and hippocampal oscillations; the incompletely explained methods and uncertainties about how some methods/illustrations relate to results with occasional apparent mismatches; and statistical questions, including the validity of interpretations given the relatively small numbers of mice. Since many of the comments stem from the difficulty the reviewers had in following what the experiments where and why certain approaches were taken as being valid, the reviewers agreed that there may be multiple appropriate ways in which you may choose to address the comments. We are therefore not stipulating precisely what the explicit experiments, analyses, and or revisions should be. Nevertheless, for the revision, please address the concerns by making the following essential revisions:

1) Acknowledging the limitations of the correlative data address/explain more clearly the link between cerebellar theta and hippocampal oscillations;

2) Providing a clearer explanation of methods and justification of their validity (especially regarding temporal components of manipulations) and ensure that the text and interpretations are consistent with the data (as illustrated and as summarized);

3) Addressing the statistical queries in the reviewers' comments, including addressing the small numbers of mice on what conclusions were based in some experiments.

The specific (concatenated) comments in the reviewers' words, with editorial notes about which essential revision(s) they primarily pertain to in brackets, are given below to help guide you in your revision. Some points span multiple categories of revisions, but the indicators are given to try to provide some structure and clarity. Also, the full comments are included for completeness, but because some items were noted by more than one reviewer, some redundancy is present. We realize that in some cases a single response may answer more than one specific comment, which is fine, as long as all points are addressed in the revision.

The general assessments are also provided.

General assessments:

*Reviewer #1:*

This is an interesting paper that uses (1) anatomy to demonstrate that topographically restricted regions of the cerebellar cortex are connected to the hippocampus and (2) that local field potential measured oscillatory activity in the theta band (6 – 12 Hz) is coordinated between sub-regions of the cerebellum and the hippocampus during navigation. I find the topic timely and the results of broad interest but do have some concerns regarding the robustness of the effects and the limited presence of oscillatory activity in the theta frequency range in the cerebellum. The report is also primarily correlative in nature – although there are interesting novel insights (the topographic nature of cerebellar-hippocampal connectivity and the increase in coherence between specific cerebellar regions and the hippocampus with navigation/experience).

*Reviewer #2:*

This is an interesting study describing the interactions between cerebellum and hippocampus during a goal-directed behavior. To this end the authors first studied the topography of the cerebellum regions projecting to the hippocampus using rabies via a polysynaptic (disynaptic?) pathway. Then, they demonstrate a spectral coherence of the theta oscillations (an important neural mechanism that supports neural communication) between these regions during active movement in the home cage and during learning of a goal-directed behavior task in a linear track and in virtual reality conditions in mice; difference of coherence between the regions seem to match the density of connections, and changes of coherence intensity takes place during learning in freely moving mice but not in virtual reality. The overall results seem appealing for a broad readership interested in functional relationship between different structures, here the cerebellum and the hippocampus during behavior, although the cause of temporal variations in coherence remains elusive (caused by upstream gating of theta-modulated inputs? changes in cerebellar oscillation amplification?) and is not discussed. Cellular recordings in the cerebellum showing phase-locking to hippocampal theta oscillations to ascertain that LFP is reflecting intra-cerebellar activity (and not simply external entrainment) would have been helpful but this would require more than can be completed in a reasonable revision time. Anyway, this work could be much improved by the authors with new analysis and clarification of interpretation of their data.

*Reviewer #3:*

This is a fascinating manuscript in which the authors provide anatomical and physiological evidence for cerebellar interactions with the hippocampus. Based on the data presented, it is clear that regions of the cerebellar vermis and hemisphere, as well as their associated deep nuclei are a source of multisynaptic drive to the hippocampus. The anatomical results provide the most compelling data so far on this issue. Overall, I find the results believable, but requiring additional documentation. For example, the injection sites in the hippocampus are not clearly illustrated. The neurons that mediate the transneuronal transport have not been identified. It is not clear whether the connection between the cerebellar nuclei and the hippocampus is mediated by one or two neurons. The neurons that mediate this connection do not appear to have been analysed. At least the authors have neither presented nor illustrated this critical information. Thus, we are left with clear evidence that multiple regions of the cerebellum influence the hippocampus, but the subcortical or cortical regions that mediate this influence are uncertain. I believe that the authors should analyse and illustrate this critical information prior to publication.

Major points [concatenated from all reviewers]:

1) [Related to Essential Point 1] The authors indicate that transient oscillatory activity was observed in the 6-12 Hz range in the cerebellum. To navigate, sensory-motor information must be integrated nearly continuously and how the cerebellum meaningful contributes to this calculation in the hippocampus is not clear to me given the transient nature of theta in the cerebellum.

2) [Related to Essential Point 1] In addition, more information should be presented regarding this feature [the transience] of cerebellar theta. For example, what is the frequency or duration of transient theta oscillations in the home cage versus various tasks? Are the transitions between theta frequency and other frequencies discrete (i.e. what are the dynamics of preceding and following activity in the theta frequency range?). It's also not always clear whether analyses are focusing on identified transient periods of theta or longer periods of theta. It's possible I missed this information but it should be clear in the main text.

3) [Related to Essential Point 1] The authors show only power spectra and coherence until 40 Hz and they focused on theta but there is clearly increased activity in beta frequencies which are not commented nor discussed (while task-related changes in beta frequencies have also been reported in the hippocampus). Recording were performed with sampling frequency of 1kHz (from Materials and methods) so the authors could also analyze higher frequencies up to 200 Hz. A large fraction of the gamma band is currently not really studied.

4) [Related to Essential Point 1] In Figure 4G-H, "observed changes restricted to the theta band": around 30Hz there is definitely a peak, both in trial 1 and in trial 20 (even larger/higher in trial 20). A shift in 30Hz peak can also be seen on the z-score power of the HPC (Figure 4E and F). Notably, this peak seems absent from the homecage recordings (Figure 3D). In virtual reality, this peak at 30 Hz seems absent, except when selecting specific epochs comparable to the linear track trials (Figure 7C). There might be something to investigate here for the authors, please check the absence of significance in this frequency band.

5) [Related to Essential Point 1] The coherence analysis is classically difficult to interpret because of volume conduction. Indeed, the authors did not find a clear peak of theta in power spectrum in the cerebellum but only transient oscillations in traces. The authors make an argument on distance, but this may be misleading if the dipolar nature of the source is not taken into account. Imaginary part of coherency is robust against volume conduction and should be used to replicate the findings (see Nolte et al., 2004).

6) [Related to Essential Point 1] The comparison of freely moving and virtual-reality is tricky. The learning curves look very different in Figure 4B and Figure 6C. Please provide a comparison between learning curves between linear track and virtual reality conditions. This impact on the choice of the epochs with similar behavior performances to compare in Figure 7. How similar is this behavior? Is it simply running speed? The VR animals seem overtrained while the freely moving seem to be learning during the session; this may result in differences of hippocampal engagement, beyond differences of sensory context.

7) [Related to Essential Point 2] It is a bit hard to make sense of the time-lapse of the cerebellum; from the Figure 1—figure supplement 2, it seems that the cerebellar nuclei is infected 18 hours after the primary infections observed in the structures which project to the cerebellum, suggesting that the cerebellum is one synapse upstream these hippocampus-projecting regions. Is any of the structures labelled at 30h known to receive inputs from the cerebellar nuclei? Alternatively, is it conceivable that some infection took place in the cerebral cortex overlying the hippocampus? The size of the pipette used for the injection is not provided. A picture of the cortex over the infection site would be useful for the reader to judge of the risk of virus leaks to the cortex.

8) [Related to Essential Point 2] Regarding the anatomical study in subsection “A precise topography of the cerebellum regions projecting to the hippocampus”, second paragraph, the authors stated that the presence of labeling in the contralateral hippocampus at the earliest survival time is at 30h vs. 48h for only contralateral markings in cerebellar output nuclei DCN: that's more than the 12h required for an infection cycle, so they should start seeing cells as well in the ipsilateral cerebellar nuclei, no? The time lapse of the retroinfection isn't very clear: both hemispheres of the cerebellar cortex are infected at the same moment, while the DCNs are infected at different times (cf. Supplementary Figure 2)? The symmetry in the pattern of infection in the cerebellar cortex suggests connections going to the same area, but the timelapse of the infection installs some doubts. Please clarify.

9) [Related to Essential Point 2] I do not see some of the findings mentioned in the following text illustrated in Figure 1. "Rather, RABV/CTb-labeled neurons were found in two well described subcortical pathways leading to the DG of the hippocampus (first cycle of infection). One labeled pathway included the diagonal band of Broca and the septum. The other labeled hippocampal input pathway included the lateral entorhinal and perirhinal cortices (Figure 1—figure supplement 3)(Mosko et al., 1973; Dolorfo and Amaral, 1998; Witter, 2007.… "

10) [Related to Essential Point 2] While the tracing was performed by injections in the dentate gyrus, recordings were performed in another region, the CA1 area. Please clarify the motivation of these choices and discuss how this may impact the result.

11) [Related to Essential Point 2] Subsection “A precise topography of the cerebellum regions projecting to the hippocampus”, last paragraph: Several pieces of anatomical information are missing from the analysis.

a) What is the location of first order neurons in subcortical structures that are labeled by retrograde transport after hippocampal injections of RV?

b) What is the time course of the appearance of this first order labeling?

c) What is the time course of the appearance of neurons in fastigial, interpositus and dentate? Do they all appear at the same time in individual animals? Is the timing of their appearance fully consistent with all the neurons in the deep cerebellar nuclei being second order neurons labeled by retrograde transneuronal transport? Or is it possible that some of the neurons are third order neurons? In other words, do neurons in the fastigial nucleus label 8 hours before neurons in the dentate? If so, then this would be consistent with fastigial neurons being second order and dentate neurons being third order.

12) [Related to Essential Point 2] Is the time course of the first labeled neurons in the deep nuclei fully consistent with their being second order neurons? Or is it possible that even the earliest labeled neurons are third order neurons that are labeled by transneuronal transport through two subcortical neurons?

13) [Related to Essential Point 2] What is the time course of labeled Purkinje cells in different regions of the cerebellar cortex? Do neurons in the vermis label ~8 hours prior to neurons in the hemisphere?

14) [Related to Essential Point 2] As the anatomical data currently stand, their analysis and/or presentation are incomplete. The following statement is neither clearly stated in the Results section nor clearly illustrated. "Notably, all of these regions contained RABV+ cells at 48h p.i., and thus they cannot be excluded as potential routes towards the hippocampus."

15) [Related to Essential Point 2 and 3] Some additional methods are needed. It was not clear to me how recordings were consistently selected for inclusion in various analysis. For example, the last paragraph of the subsection “Cerebello-hippocampal physiological interactions in a familiar home-cage environment”, lists 23 values from 13 mice. How were these sessions selected? Which mice did these sessions come from (up to two sessions per mouse?) and why would session numbers used in analyses differ between mice? This is just one example but the comment applies to statistics throughout the paper.

16) [Related to Essential Point 3] The authors provided a measure of physiological interaction – namely theta oscillations between cerebellum and hippocampus. First they assessed the cerebello-hippocampal interactions with coherence in the home cage and then during the learning of a goal directed behavior in a linear track and in virtual reality conditions. In the third paragraph of the subsection “Cerebello-hippocampal interactions during the learning of a goal-directed behavior” and Figure 4E-F: The shift in mean theta power and peak frequency is reported significant in the text but not clear on the figure when no visible difference in the theta peak between trial 1 and 20 was observed. Also when looking at the numbers in this z-score power (subsection “Cerebello-hippocampal interactions during the learning of a goal-directed behavior”, third paragraph), the difference is barely visible and the SEM give overlapping numbers (1st trial between 1.59 and 1.69, 20th trial between 1.68 and 1.76 for example): the statistical test seems to be conflicting with these results. Please clarify.

17) [Related to Essential Point 3] Some of the effects appear to rely heavily on smaller numbers of mice, although the authors do have quantitative explanations for the variability they observe (i.e. the recording distance from the midline in Figure 3F). Even so, Figure 3F seems to be primarily driven by two mice and in Figure 5, I was unsure of how the authors interpreted the high degree of variability in mean coherence in the Crus I data set.

18) [Related to Essential Point 3] The authors recorded the activity in both left and right hippocampus in all mice and they noted no difference (subsection “Cerebello-hippocampal physiological interactions in a familiar home-cage environment”, fourth paragraph) in the analysis of cerebello-hippocampal coherence. However, in Figure 4 to Figure 7 the data from different animals are indifferentially reported from data from the same animal. Indeed, the significant effects, for example for Crus I in Figure 5 and Figure 7D may be only due to a few values, possibly coming from only one or two animals; moreover it seems that some of the statistics are not performed using repeated-measure procedures. Please provide a clear view on left/right recording from each animal and clarify the statistics (e.g. averaging values from single animals). Overall, the number of mice seems a bit small for the task-related coherence analysis to evaluate reproducibility of the results.

---

## [Author Response]

Essential revisions:The three reviewers all found aspects of the work to be of interest. They all, however, had fairly extensive critiques, including comments on the limitations of the correlative data and the ambiguity of the link between cerebellar and hippocampal oscillations; the incompletely explained methods and uncertainties about how some methods/illustrations relate to results with occasional apparent mismatches; and statistical questions, including the validity of interpretations given the relatively small numbers of mice. Since many of the comments stem from the difficulty the reviewers had in following what the experiments where and why certain approaches were taken as being valid, the reviewers agreed that there may be multiple appropriate ways in which you may choose to address the comments. We are therefore not stipulating precisely what the explicit experiments, analyses, and or revisions should be. Nevertheless, for the revision, please address the concerns by making the following essential revisions:1) Acknowledging the limitations of the correlative data address/explain more clearly the link between cerebellar theta and hippocampal oscillations;2) Providing a clearer explanation of methods and justification of their validity (especially regarding temporal components of manipulations) and ensure that the text and interpretations are consistent with the data (as illustrated and as summarized);

This essential point 2 concerns the anatomical part of the manuscript. Following reviewer’s suggestions we have now illustrated the injection site in more detail and also performed a detailed analysis of the neurons that mediate the transneuronal transport. To provide a detailed documentation of these points, Figure 1 now has 6 associated supplementary figures:

- Figure 1—figure supplement 1 illustrates the injection site for all the mice. We also illustrated this injection site with CTB labeling pictures. Following this new analysis we have removed one 30h animal due to leaks into the cortex.

- Figure 1—figure supplement 2 corresponds to RABV labeling at 30h post injection.

- Figure 1—figure supplement 3 corresponds to CTB labeling. This figure confirms that the predominantly RABV labeled structures at 30h post injection are first order structures.

- Figure 1—figure supplement 4 corresponds to RABV labeling at 48h post injection. Structures labeled at 48h that were not labeled at 30h are potential second order structures (written in bold in the table)

- Figure 1—figure supplement 5 is a summary table of RABV labeling in cerebellum and vestibular nuclei at 58h post injection.

- Figure 1—figure supplement 6 illustrating the topographical distribution of DCN labeling at 66h.

3) Addressing the statistical queries in the reviewers' comments, including addressing the small numbers of mice on what conclusions were based in some experiments.The specific (concatenated) comments in the reviewers' words, with editorial notes about which essential revision(s) they primarily pertain to in brackets, are given below to help guide you in your revision. Some points span multiple categories of revisions, but the indicators are given to try to provide some structure and clarity. Also, the full comments are included for completeness, but because some items were noted by more than one reviewer, some redundancy is present. We realize that in some cases a single response may answer more than one specific comment, which is fine, as long as all points are addressed in the revision.[…]1) [Related to Essential Point 1] The authors indicate that transient oscillatory activity was observed in the 6-12 Hz range in the cerebellum. To navigate, sensory-motor information must be integrated nearly continuously and how the cerebellum meaningful contributes to this calculation in the hippocampus is not clear to me given the transient nature of theta in the cerebellum.

We agree with the reviewer that understanding the temporal dynamics of cerebellum and hippocampus activity is a major question. Therefore, our spectrogram and coherogram analyses have been adapted to use parameters that allow for better temporal resolution (1s window in 0.1s steps compared with the previous 10s with 1s step). With these parameters we were able to explore temporal and spatial dynamics at the level of single trials.

In the linear track condition we have a constrained behavior and the task can be divided into individual trials (each run from one end of the corridor to the next goal zone) inside a session (each 12 minute block). We have now analysed the spectrogram averaged by the distance to the reward in late training and we observed rather continuous theta band activity during the whole goal-directed behavior (see new Figure 4Ei). We also computed coherograms averaged by the distance to the reward. Our new analysis reveals sustained coherence between hippocampus and Crus I, but not Lob II/III or Lob VI, in the theta band (see new Figure 4Eii, Eiii, 4H). Interestingly, activity and synchronization in this frequency band is also absent when analyzing trial 1 (new Figure 4D, F, H).

These results (subsection “Cerebello-hippocampal interactions during the learning of a goal-directed behavior”) are in line with the general findings described in the previous version of the manuscript, i.e. the increase of coherence between the Crus I region and the hippocampus when the animal performs a goal directed behavior (Figure 4C), and offer further insights on the dynamics of this interaction during ongoing behavior.

In the homecage, the animal performed a mixture of different behavioral states that are difficult to separate or divide on a “trial by trial” basis. To analyze the nature of the theta activity in the home cage, we looked at the distribution of instantaneous theta power. We found a skewed but unimodal distribution arguing against the existence of two differentiated states (presence versus absence of theta activity) (new Figure 3D) (see also new Results subsection “Cerebello-hippocampal physiological interactions in a familiar home-cage environment”).

Altogether, these new analyses sustain the idea that the cerebellum may provide the hippocampus with integrated sensory-motor information within the contextual framework of the goal directed behavior (see new Discussion section, eighth paragraph), which should occur in a nearly continuous manner (as pointed out by the reviewer).

2) [Related to Essential Point 1] In addition, more information should be presented regarding this feature [the transience] of cerebellar theta. For example, what is the frequency or duration of transient theta oscillations in the home cage versus various tasks? Are the transitions between theta frequency and other frequencies discrete (i.e. what are the dynamics of preceding and following activity in the theta frequency range?). It's also not always clear whether analyses are focusing on identified transient periods of theta or longer periods of theta. It's possible I missed this information but it should be clear in the main text.

As discussed in point 1, we now show evidence for a continuum in the cerebellar theta activity rather than transient activity. Briefly, across the all conditions, we analysed LFP power and coherence during epochs of active movement (speed above 3 cm/s). Furthermore, for the plots in Figure 4F-H we have averaged the spectra and coherence within the range of distances from the reward that showed significant differences across combinations in session 20 (-60 to -20 cm, see Figure 4Eiii).

3) [Related to Essential Point 1] The authors show only power spectra and coherence until 40 Hz and they focused on theta but there is clearly increased activity in beta frequencies which are not commented nor discussed (while task-related changes in beta frequencies have also been reported in the hippocampus). Recording were performed with sampling frequency of 1kHz (from Materials and methods) so the authors could also analyze higher frequencies up to 200 Hz. A large fraction of the gamma band is currently not really studied.

We have now analysed a much larger frequency profile (2-300 Hz) for both our LFP power and coherence analyses. We have analysed the differences between cerebello-hippocampal combinations in the frequency bands theta (6-12 Hz), beta (13-29 Hz) and low-gamma (30-48 Hz) and then used a two-way repeated measures ANOVA for frequency band and combination factors. In all conditions the only significant values or changes across combinations were found within the theta range (new Figures 3C, F, I; 4D-H) (see below for detailed statistical analysis and Results section).

Homecage:

Frequency band effect F_2,76_ = 22.42, p < 0.0001

Combination effect F_2,38_ = 2.843, p = 0.0707 Interaction effect, F_4,76_ = 3.825, p = 0.0069

FDR corrected multiple comparisons between combinations for each frequency band:

Theta:

Lob VI-HPC vs. Crus I-HPC, corrected p = 0.003

Lob VI-HPC vs. Lob II/III-HPC, corrected p < 0.0001

Crus I-HPC vs. Lob II/III-HPC, corrected p = 0.0964

Βeta:

Lob VI-HPC vs. Crus I-HPC, corrected p = 0.8314

Lob VI-HPC vs. Lob II/III-HPC, corrected p = 0.8314

Crus I-HPC vs. Lob II/III-HPC, corrected p = 0.8314

Gamma:

Lob VI-HPC vs. Crus I-HPC, corrected p = 0.7015

Lob VI-HPC vs. Lob II/III-HPC, corrected p = 0.5968

Crus I-HPC vs. Lob II/III-HPC, corrected p = 0.5968

Linear track:

Session 1:

Frequency band effect F_2,26_ = 9.283, p = 0.0009

Combination effect F_2,13_ = 2.193, p = 0.1512

Interaction effect, F_4,26_ = 2.044, p = 0.1176

Session 20:

Frequency band effect F_2,26_ = 13.42, p < 0.0001

Combination effect F_2,13_ = 6.545, p = 0.0108 Interaction effect, F_4,26_ = 5.242, p = 0.0031

FDR corrected multiple comparisons between combinations for each frequency band:

Theta:

Lob VI-HPC vs. Crus I-HPC, corrected p = 0.0016

Lob VI-HPC vs. Lob II/III-HPC, corrected p = 0.0178

Crus I-HPC vs. Lob II/III-HPC, corrected p < 0.0001

Beta:

Lob VI-HPC vs. Crus I-HPC, corrected p = 0.4051

Lob VI-HPC vs. Lob II/III-HPC, corrected p = 0.9882

Crus I-HPC vs. Lob II/III-HPC, corrected p = 0.3978

Gamma:

Lob VI-HPC vs. Crus I-HPC, corrected p = 0.5312

Lob VI-HPC vs. Lob II/III-HPC, corrected p = 08208

Crus I-HPC vs. Lob II/III-HPC, corrected p = 0.4084

We have now also modified our plots (new Figure 3C, F and Figure 4F and G) to a logarithmic scale which to display the full frequency range (2-300 Hz).

4) [Related to Essential Point 1] In Figure 4G-H, "observed changes restricted to the theta band": around 30Hz there is definitely a peak, both in trial 1 and in trial 20 (even larger/higher in trial 20). A shift in 30Hz peak can also be seen on the z-score power of the HPC (Figure 4E and F). Notably, this peak seems absent from the homecage recordings (Figure 3D). In virtual reality, this peak at 30 Hz seems absent, except when selecting specific epochs comparable to the linear track trials (Figure 7C). There might be something to investigate here for the authors, please check the absence of significance in this frequency band.

As mentioned in point 3 (above), we have analysed the whole frequency profile. The significant difference is restricted to the theta band (6-12 Hz, see statistics above).

5) [Related to Essential Point 1] The coherence analysis is classically difficult to interpret because of volume conduction. Indeed, the authors did not find a clear peak of theta in power spectrum in the cerebellum but only transient oscillations in traces. The authors make an argument on distance, but this may be misleading if the dipolar nature of the source is not taken into account. Imaginary part of coherency is robust against volume conduction and should be used to replicate the findings (see Nolte et al., 2004).

Whilst we accept that volume conduction is indeed an inherent caveat of coherence analysis, we have made many efforts in our approach to minimize its impact on our findings. Distance between recording sites is not our only control and we would like to list them below (see also Discussion section) as well as describing the two new analyses we have now performed.

1) Rather than using a common reference electrode, our recordings were bipolar, with each recording electrode being locally and independently referenced. This method has been described as being an effective method in reducing the general reference electrode issue (Kajikawa and Schroeder, 2011).

2) By recording simultaneously from hippocampus and multiple cerebellar regions we have been able to demonstrate that the observed coherence is non-homogenous among the different cerebellar lobules in contrast to what one would expect if theta was volume conducted from a common location.

In addition to these previous points and following reviewer’s comments and suggestions, we now have calculated the imaginary coherence and performed single cell recordings from the cerebellum.

3) The findings described in Figure 4 are replicated with high similarity when imaginary coherence is calculated (see new Figure 4—figure supplement 3), again suggesting that the described coherence is not generated by volume conduction.

4) Finally, we have conducted a new series of technically challenging experiments in which we have recorded single-unit activity in the cerebellar cortex alongside with LFP in the hippocampus (see new Figure 3—figure supplement 4) of head fixed L7-ChR2 mice. We took advantage of this mouse line to allow positive photo-identification of cerebellar Purkinje cells during recordings (determined by cell responses to blue light illumination; Cf Chaumont et al., 2013) (Figure 3—figure supplement 3). Interestingly, we found that 16 of 22 cells (6 mice) recorded in cerebellar lobule VI showed a robust response during optical illumination. Of these 16 cells, almost one-third (31%) were found to be significantly phase-locked to hippocampal theta oscillations. This finding also argues against a volume conduction origin of the coherence described in this manuscript (see new Results section).

6) [Related to Essential Point 1] The comparison of freely moving and virtual-reality is tricky. The learning curves look very different in Figure 4B and Figure 6C. Please provide a comparison between learning curves between linear track and virtual reality conditions. This impact on the choice of the epochs with similar behavior performances to compare in Figure 7. How similar is this behavior? Is it simply running speed? The VR animals seem overtrained while the freely moving seem to be learning during the session; this may result in differences of hippocampal engagement, beyond differences of sensory context.

Following reviewer’s suggestion we performed an individual analysis of the behaviour of each mouse during the virtual reality training. Among the 6 trained mice, two improved their performances and four remained with stable performances. Therefore, we now have analysed these mice separately (see Figure 4—figure supplement 4). The different examples show different patterns of activity and coherence. Interestingly, as it was the case during the linear track training, coherent activity between Crus I and hippocampus appeared when mice learned the task.

7) [Related to Essential Point 2] It is a bit hard to make sense of the time-lapse of the cerebellum; from the Figure 1—figure supplement 2, it seems that the cerebellar nuclei is infected 18 hours after the primary infections observed in the structures which project to the cerebellum, suggesting that the cerebellum is one synapse upstream these hippocampus-projecting regions. Is any of the structures labelled at 30h known to receive inputs from the cerebellar nuclei?

The fact that at 48h, labeled cells in the DCN are sparsely located and their processes weakly stained could suggest the beginning of a third order infection cycle and thus tends to suggest that the main infection of the DCN starts at 58 h post infection and involves two relays to reach the hippocampus (Figure 1—figure supplement 4).

However, at 30h post infection, staining is mainly found in the medial septum diagonal band of Broca (MSDB), the lateral and medial entorhinal cortices and the perirhinal cortex as well as the supramammillary nucleus (SUM) (Figure 1—figure supplement 2). We now confirm that these structures correspond to first order as CTB staining was also systematically found in the same structures (see Figure 1—figure supplement 3).

Among these structures, the septum has been described to receive direct projection from the fastigial nucleus in cats (Paul et al., 1973). In addition, we also found, in two mice out of four, RABV staining in the hypothalamus (including SUM in three animals) and raphe nucleus at 30h as well as CTB staining (see below). As these regions receive direct projections from the DCN (Teune et al., 2000), the few cells observed in the DCN 48h post infection could be related to the hypothalamic (potentially including the supramammillary nuclei (SUM)), and/or raphe nucleus staining observed at 30h.

Altogether, these results suggest multiple convergent pathways from the DCN to the hippocampus. Single relay pathways could be envisioned through the septum, the hypothalamus (potentially including the supramammillary bodies (SUM)) and the raphe nucleus (see new Results and Discussion sections as well as new supplementary figures associated with Figure 1). Other pathways including 2 relays through either the lateral and medial entorhinal cortex and/or the perirhinal cortex are also possible.

Alternatively, is it conceivable that some infection took place in the cerebral cortex overlying the hippocampus? The size of the pipette used for the injection is not provided. A picture of the cortex over the infection site would be useful for the reader to judge of the risk of virus leaks to the cortex.

The size of the pipette diameter is around 200 µm. We have now added this information in the Materials and methods section. Please note that in the majority of the cases, the cortex overlying the hippocampus has been severely damaged during the injection, which renders a possible leak very unlikely. Nevertheless, in the other cases, no virus leak was detected in the cortex overlying the hippocampus. We have now added a supplementary figure to illustrate the cortex over the infection site (see Figure 1—figure supplement 1E).

8) [Related to Essential Point 2] Regarding the anatomical study in subsection “A precise topography of the cerebellum regions projecting to the hippocampus”, second paragraph, the authors stated that the presence of labeling in the contralateral hippocampus at the earliest survival time is at 30h vs. 48h for only contralateral markings in cerebellar output nuclei DCN: that's more than the 12h required for an infection cycle, so they should start seeing cells as well in the ipsilateral cerebellar nuclei, no? The time lapse of the retroinfection isn't very clear: both hemispheres of the cerebellar cortex are infected at the same moment, while the DCNs are infected at different times (cf. Supplementary Figure 2)? The symmetry in the pattern of infection in the cerebellar cortex suggests connections going to the same area, but the timelapse of the infection installs some doubts. Please clarify.

The labeling at 48h in the DCN is too sparse to allow any quantitative comparison between ipsi and contra DCN (see Figure 1—figure supplement 3). At 58h, the labeling in the DCN is more abundant and stronger. At this time course, we did not find any difference between ipsi and contralateral staining in fastigial and dentate nuclei. The labeling in the interpositus remained weak in both sides (Figure 1—figure supplement 6).

9) [Related to Essential Point 2] I do not see some of the findings mentioned in the following text illustrated in Figure 1. "Rather, RABV/CTb-labeled neurons were found in two well described subcortical pathways leading to the DG of the hippocampus (first cycle of infection). One labeled pathway included the diagonal band of Broca and the septum. The other labeled hippocampal input pathway included the lateral entorhinal and perirhinal cortices (Figure 1—figure supplement 3)(Mosko et al., 1973; Dolorfo and Amaral, 1998; Witter, 2007.… "

Indeed as mentioned by the reviewer, the CTB labelling was not shown in the previous version of the manuscript. We have now added CTB labeling corresponding to the first order labelled neurons (new Figure 1—figure supplement 3). We also replaced the table (previous Figure 1—figure supplement 2) by the different supplementary figures mentioned above. We have therefore replaced the sentence by a better and more detailed description of the structures labelled at the different survival time (see subsection “Cerebellar projections to the hippocampus are precisely topographically organized”).

10) [Related to Essential Point 2] While the tracing was performed by injections in the dentate gyrus, recordings were performed in another region, the CA1 area. Please clarify the motivation of these choices and discuss how this may impact the result.

We chose to inject into the dentate gyrus for two reasons: 1) to target the maximal number of potential afferent inputs of the hippocampus and 2) we wanted to minimize the risk of a leak in the regions overlying the hippocampus. It is important to note that the site of viral injection involves both the stratum lacunosum-moleculare and the molecular layer of dentate gyrus which correspond to the main axonal entrance to CA1 and DG. The LFP recording was centered on CA1 since we previously described the influence of cerebellar plasticity on CA1 neuronal activity (Rochefort et al., 2011). We now have added a paragraph in the Results section of the manuscript to add these precisions.

11) [Related to Essential Point 2] Subsection “A precise topography of the cerebellum regions projecting to the hippocampus”, last paragraph: Several pieces of anatomical information are missing from the analysis.a) What is the location of first order neurons in subcortical structures that are labeled by retrograde transport after hippocampal injections of RV?

We now present two supplementary figures describing all the regions containing first order neurons, 30h after hippocampal infection by rabies virus (new Figure 1—figure supplement 2) and confirmed by CTB (see new Figure 1—figure supplement 3). The main subcortical regions infected at 30h and thus projecting directly to the hippocampus are the Medial septum and diagonal band of Broca (MSDB), the lateral and medial entorhinal as well as the perirhinal cortices and the supramammillary bodies (see also answer to point 7 and 9).

b) What is the time course of the appearance of this first order labeling?

In our manuscript, the time course used to label first order neurons is 30h. We did not test a shorter survival time since this time is classically used in rodents tracing studies (e.g. Coulon et al., 2011) and allows visualization of a maximal number of directly projecting structures towards the injection site with a good intensity of labeling as confirmed by CTB labeling.

c) What is the time course of the appearance of neurons in fastigial, interpositus and dentate? Do they all appear at the same time in individual animals? Is the timing of their appearance fully consistent with all the neurons in the deep cerebellar nuclei being second order neurons labeled by retrograde transneuronal transport? Or is it possible that some of the neurons are third order neurons? In other words, do neurons in the fastigial nucleus label 8 hours before neurons in the dentate? If so, then this would be consistent with fastigial neurons being second order and dentate neurons being third order.

The labeling at 48h in the DCN is too sparse to allow any quantitative comparison between ipsilateral and contralateral DCN (see Figure 1—figure supplement 3). At 58h, the labeling in the DCN is more abundant and stronger. At this time course, we did not find any difference between ipsi and contralateral staining in fastigial and dentate nuclei. The labeling in the interpositus remained weak in both sides (see Figure 1—source data 1).

12) [Related to Essential Point 2] Is the time course of the first labeled neurons in the deep nuclei fully consistent with their being second order neurons? Or is it possible that even the earliest labeled neurons are third order neurons that are labeled by transneuronal transport through two subcortical neurons?

According to CTB staining, we consider that 30h post-infection stained cells represent first order neurons and that 48h post-infection stained cells form second order neurons. It is thus possible that the few labeled neurons observed in the DCN at 48h post hippocampal infection represent second order neurons (see a detailed answer to this point in point 7 and in the Results section of the manuscript).

13) [Related to Essential Point 2] What is the time course of labeled purkinje cells in different regions of the cerebellar cortex? Do neurons in the vermis label ~8 hours prior to neurons in the hemisphere?

Labeled neurons were found in both the hemisphere and the vermis at 58h post infection (see new Figure 1—figure supplement 5).

14) [Related to Essential Point 2] As the anatomical data currently stand, their analysis and/or presentation are incomplete. The following statement is neither clearly stated in the Results section nor clearly illustrated. "Notably, all of these regions contained RABV+ cells at 48h p.i., and thus they cannot be excluded as potential routes towards the hippocampus."

The revised manuscript now contains a paragraph in the Results section presenting the different structures stained at 48h. The table (Figure 1—figure supplement 4) describing these regions is also updated. Interestingly, among the different regions labeled at 48h post infection, several midbrain and pontine regions such as the PAG, the Nucleus Incertus and the LtDG are known to receive direct projections from the DCN and could therefore represent putative relays between DCN and hippocampus.

15) [Related to Essential Point 2 and 3] Some additional methods are needed. It was not clear to me how recordings were consistently selected for inclusion in various analysis. For example, the last paragraph of the subsection “Cerebello-hippocampal physiological interactions in a familiar homecage environment”, lists 23 values from 13 mice. How were these sessions selected? Which mice did these sessions come from (up to two sessions per mouse?) and why would session numbers used in analyses differ between mice? This is just one example but the comment applies to statistics throughout the paper.

We apologized if the data used for the different analysis were not clear enough in the previous version of the manuscript. The difference between values and animals referred here it was due to the use of HPCright – cerebellar coherence and HPCleft – cerebellar coherence values from the same animal when both hippocampal electrodes were on target (after histological verification). In order to clarify this and reduce the potential impact of this pooling on the statistics we have now calculated and used the mean between these two values so each animal is represented only once. A detailed explanation has been added in the Results section of the new version of the manuscript.

16) [Related to Essential Point 3] The authors provided a measure of physiological interaction – namely theta oscillations between cerebellum and hippocampus. First they assessed the cerebello-hippocampal interactions with coherence in the home cage and then during the learning of a goal directed behavior in a linear track and in virtual reality conditions. In the third paragraph of the subsection “Cerebello-hippocampal interactions during the learning of a goal-directed behavior” and Figure 4E-F: The shift in mean theta power and peak frequency is reported significant in the text but not clear on the figure when no visible difference in the theta peak between trial 1 and 20 was observed. Also when looking at the numbers in this z-score power (subsection “Cerebello-hippocampal interactions during the learning of a goal-directed behavior”, third paragraph), the difference is barely visible and the SEM give overlapping numbers (1st trial between 1.59 and 1.69, 20th trial between 1.68 and 1.76 for example): the statistical test seems to be conflicting with these results. Please clarify.

Following our extensive re-analysis, the figures referred to by the reviewer are no longer present in the manuscript. We have endeavoured to ensure that all new analysis descriptions, statistics and results are clear to the reader. Indeed, changes in power are now better visualised after the trial by trial analysis averaged by distance from reward (Figure 4D-G). We have added information relating to the new findings in the Results section.

17) [Related to Essential Point 3] Some of the effects appear to rely heavily on smaller numbers of mice, although the authors do have quantitative explanations for the variability they observe (i.e. the recording distance from the midline in Figure 3F). Even so, Figure 3F seems to be primarily driven by two mice and in Figure 5, I was unsure of how the authors interpreted the high degree of variability in mean coherence in the Crus I data set.

Following re-analysis, we have now averaged coherence values obtained between the cerebellum and left/right hippocampus. Figure 3F (now Figure 3J) shows a strong correlation between theta coherence and electrode positioning within lobule VI. This correlation remains robust after our updated analysis in which we averaged down our coherence values so each data point corresponds to one animal. In terms of Crus I coherence variations, we have also conducted extensive reanalysis of this data set and we find that 3/4 animals show similar patterns of Crus I coherence change in the linear track task (Figures 4H and 5). Variations in recording electrode position within Crus I may have also contributed to variations in coherence levels as for lobule VI, but we were unable to accurately reconstruct the electrode positions with high enough resolution to systematically check this as we did for lobule VI.

18) [Related to Essential Point 3] The authors recorded the activity in both left and right hippocampus in all mice and they noted no difference (subsection “Cerebello-hippocampal physiological interactions in a familiar home-cage environment”, fourth paragraph) in the analysis of cerebello-hippocampal coherence. However, in Figure 4 to Figure 7 the data from different animals are indifferentially reported from data from the same animal. Indeed, the significant effects, for example for Crus I in Figure 5 and Figure 7D may be only due to a few values, possibly coming from only one or two animals; moreover it seems that some of the statistics are not performed using repeated-measure procedures. Please provide a clear view on left/right recording from each animal and clarify the statistics (e.g. averaging values from single animals). Overall, the number of mice seems a bit small for the task-related coherence analysis to evaluate reproducibility of the results.

As the reviewer points out, we observed no significant difference in values of coherence obtained between cerebellum and left or right hippocampus. The detailed comparison of HPC left vs. HPC right for each condition can be found in Figure 3—figure supplement 2 and Figure 4—figure supplement 1. Therefore, we have now changed the main figures and the statistical analysis presented in the manuscript to a single value of coherence between hippocampus and cerebellar regions per animal reflecting the average between left and right measurements when both are available.

Regarding the repeated-measure procedures they have been used when the data allowed for it; however, if this comment refers to its employment when comparing differences across combinations in a single condition (Figure 3 and Figure 4), each recording has been considered independent as they were locally referenced.